# Learned Robust PCA: A Scalable Deep Unfolding Approach for High-Dimensional Outlier Detection

**HanQin Cai**[*]
Department of Mathematics
University of California, Los Angeles
Los Angeles, CA 90095, USA
hqcai@math.ucla.edu

**Jialin Liu**
Decision Intelligence Lab
Damo Academy, Alibaba US
Bellevue, WA 98004, USA
jialin.liu@alibaba-inc.com

**Wotao Yin**
Decision Intelligence Lab
Damo Academy, Alibaba US
Bellevue, WA 98004, USA
wotao.yin@alibaba-inc.com

## Abstract

Robust principal component analysis (RPCA) is a critical tool in modern machine learning, which detects outliers in the task of low-rank matrix reconstruction. In this paper, we propose a *scalable* and *learnable* non-convex approach for high-dimensional RPCA problems, which we call Learned Robust PCA (LRPCA). LRPCA is highly efficient, and its free parameters can be effectively learned to optimize via deep unfolding. Moreover, we extend deep unfolding from finite iterations to infinite iterations via a novel feedforward-recurrent-mixed neural network model. We establish the recovery guarantee of LRPCA under mild assumptions for RPCA. Numerical experiments show that LRPCA outperforms the state-of-the-art RPCA algorithms, such as ScaledGD and AltProj, on both synthetic datasets and real-world applications.

## 1 Introduction

Over the last decade, robust principal component analysis (RPCA), one of the fundamental dimension reduction techniques, has received intensive investigations from theoretical and empirical perspectives [1–13]. RPCA also plays a key role in a wide range of machine learning tasks, such as video background subtraction [14], singing-voice separation [15], face modeling [16], image alignment [17], feature identification [18], community detection [19], fault detection [20], and NMR spectrum recovery [21]. While the standard principal component analysis (PCA) is known for its high sensitivity to outliers, RPCA is designed to enhance the robustness of PCA when outliers are present. In this paper, we consider the following RPCA setting: given an observed corrupted data matrix

$$Y = X_\star + S_\star \in \mathbb{R}^{n_1 \times n_2}, \tag{1}$$

where $X_\star$ is a rank-$r$ data matrix and $S_\star$ is a sparse outlier matrix, reconstruct $X_\star$ and $S_\star$ simultaneously from $Y$.

One of the main challenges of designing an RPCA algorithm is to avoid high computational costs. Inspired by deep unfolded sparse coding [22], some recent works [23–25] have successfully extended deep unfolding techniques to RPCA and achieved noticeable accelerations in certain applications.

---

[*]Authors are listed in alphabetic order. Correspondence shall be addressed to H.Q. Cai.

35th Conference on Neural Information Processing Systems (NeurIPS 2021).

Specifically, by parameterizing a classic RPCA algorithm and unfolding it as a feedforward neural network (FNN), one can improve its performance by learning the parameters of the FNN through backpropagation. Since RPCA problems of a specific application often share similar key properties (e.g., rank, incoherence, and amount of outliers), the parameters learned from training examples that share the properties can lead to superior performance. Nevertheless, all existing learning-based approaches call an expensive step of singular value thresholding (SVT) [26] at every iteration during both training and inference. SVT involves a full or truncated singular value decomposition (SVD), which costs from $\mathcal{O}(n^3)$ to $\mathcal{O}(n^2 r)$ flops[2] with some large hidden constant in front[3]. Thus, these approaches are not scalable to high-dimensional RPCA problems. Another issue is that the existing approaches only learn the parameters for a finite number of iterations. If a user targets at a specific accuracy of recovery, then the prior knowledge of the unfolded algorithm is required for choosing the number of unfolded layers. Moreover, if the user desires better accuracy later, one may have to restart the learning process to obtain parameters of the extra iterations.

We must ask the questions: "*Can one design a highly efficient and easy-to-learn method for high-dimensional RPCA problems?*" and "*Do we have to restrict ourselves to finite-iteration unfolding (or a fixed-layer neural network)?*" In this paper, we aim to answer these questions by proposing a *scalable* and *learnable* approach for high-dimensional RPCA problems, which has a flexible feedforward-recurrent-mixed neural network model for potentially infinite-iteration unfolding. Consequently, our approach can satisfy arbitrary accuracy without relearning the parameters.

**Related work.** The earlier works [1–3] for RPCA are based on the convex model:

$$\underset{\boldsymbol{X},\boldsymbol{S}}{\text{minimize}} \, \|\boldsymbol{X}\|_* + \lambda\|\boldsymbol{S}\|_1, \quad \text{subject to } \boldsymbol{Y} = \boldsymbol{X} + \boldsymbol{S}. \tag{2}$$

It has been shown that (2) guarantees exact recovery, provided $\alpha \lesssim \mathcal{O}(1/\mu r)$, a condition with an optimal order [4, 5]. Therein, $\alpha$ is the sparsity parameter of $\boldsymbol{S}_\star$ and $\mu$ is the incoherence parameter of $\boldsymbol{L}_\star$, which will be formally defined later in Assumptions 2 and 1, respectively. Problem (2) can be transformed into a semidefinite program and has a variety of dedicated methods, their per-iteration complexities are at least $\mathcal{O}(n^3)$ flops and some of them guarantee only sublinear convergence.

Later, various non-convex methods with linear convergence were proposed for RPCA: [7] uses alternating minimization, and it can tolerate outliers up to a fraction of only $\alpha \lesssim \mathcal{O}(1/\mu^{2/3}r^{2/3}n)$. [6] develops an alternating projections method (AltProj) that alternatively projects $\boldsymbol{Y} - \boldsymbol{S}$ onto the space of low-rank matrices and $\boldsymbol{Y} - \boldsymbol{X}$ onto the space of sparse matrices. AltProj costs $\mathcal{O}(n^2 r^2)$ flops and tolerates outliers up to $\alpha \lesssim \mathcal{O}(1/\mu r)$. An accelerated version of AltProj (AccAltProj) was proposed in [10], which improves the computational complexity to $\mathcal{O}(n^2 r)$ with sacrifices on the tolerance to outliers, which are allowed up to $\alpha \lesssim \mathcal{O}(1/\max\{\mu r^2 \kappa^3, \mu^{3/2} r^2 \kappa, \mu^2 r^2\})$, where $\kappa$ is the condition number of $\boldsymbol{X}_\star$. Note that the key technique used for outlier detection in AltProj and AccAltProj is employing adaptive hard-thresholding parameters.

Another line of non-convex RPCA algorithms reformulate the low-rank component as $\boldsymbol{X} = \boldsymbol{L}\boldsymbol{R}^\top$, where $\boldsymbol{L} \in \mathbb{R}^{n_1 \times r}$ and $\boldsymbol{R} \in \mathbb{R}^{n_2 \times r}$, and then performs gradient descend on $\boldsymbol{L}$ and $\boldsymbol{R}$ separately. Since the low-rank constraint of $\boldsymbol{X}$ is automatically satisfied under this reformulation, the costly step of SVD is avoided (except for one SVD during initialization). Specifically speaking, [8] uses a complicated objective function, which includes a practically unimportant balance regularization $\|\boldsymbol{L}^\top\boldsymbol{L} - \boldsymbol{R}^\top\boldsymbol{R}\|_F$. While GD costs only $\mathcal{O}(n^2 r)$ flops per iteration, its convergence rate relies highly on $\kappa$. Recently, ScaledGD [12] introduces a scaling term in gradient descend steps to remove the dependence on $\kappa$ in the convergence rate of GD. ScaledGD also drops the balance regularization and targets on a more elegant objective:

$$\underset{\boldsymbol{L}\in\mathbb{R}^{n_1 \times r}, \boldsymbol{R}\in\mathbb{R}^{n_2 \times r}, \boldsymbol{S}\in\mathbb{R}^{n_1 \times n_2}}{\text{minimize}} \, \frac{1}{2}\|\boldsymbol{L}\boldsymbol{R}^\top + \boldsymbol{S} - \boldsymbol{Y}\|_F^2, \qquad \text{subject to } \boldsymbol{S} \text{ is } \alpha\text{-sparse}, \tag{3}$$

where $\alpha$-sparsity means no more than a fraction $\alpha$ of non-zeros on every row and every column of $\boldsymbol{S}$. To enforce the sparsity constraint of $\boldsymbol{S}$, ScaledGD (and also GD) employs a sparsification operator:

$$[\mathcal{T}_{\widetilde{\alpha}}(\boldsymbol{M})]_{i,j} = \begin{cases} [\boldsymbol{M}]_{i,j}, & \text{if } |[\boldsymbol{M}]_{i,j}| \geq |[\boldsymbol{M}]_{i,:}^{(\widetilde{\alpha}n_2)}| \text{ and } |[\boldsymbol{M}]_{i,j}| \geq |[\boldsymbol{M}]_{:,j}^{(\widetilde{\alpha}n_1)}| \\ 0, & \text{otherwise} \end{cases}, \tag{4}$$

---

[2] For ease of presentation, we take $n := n_1 = n_2$ when discuss computational complexities.

[3] The hidden constant is often hundreds and sometimes thousands.

where $[\,\cdot\,]_{i,:}^{(k)}$ and $[\,\cdot\,]_{:,j}^{(k)}$ denote the $k$-th largest element in magnitude on the $i$-th row and in the $j$-th column, respectively. In other words, $\mathcal{T}_{\widetilde{\alpha}}$ keeps only the largest $\widetilde{\alpha}$ fraction of the entries on every row and in every column. The per-iteration computational complexity of ScaledGD remains $\mathcal{O}(n^2 r)$ and it converges faster on ill-conditioned problems. The tolerance to outliers of GD and ScaledGD are $\alpha \lesssim \mathcal{O}(1/\max\{\mu r^{3/2}\kappa^{3/2}, \mu r \kappa^2\})$ and $\alpha \lesssim \mathcal{O}(1/\mu r^{3/2}\kappa)$, respectively.

It is worth mentioning there exist some other RPCA settings that high-dimensional problems can be efficiently solved. For example, if the columns of $X_\star$ are independently sampled over a zero-mean multivariate Gaussian distribution, an efficient approach is using Grassmann averages [27, 28].

Deep unfolding is a technique that dates back to 2010 from a fast approximate method for LASSO: LISTA [22] parameterizes the classic Iterative Shrinkage-Thresholding Algorithm (ISTA) as a fully-connected feedforward neural network and demonstrates that the trained neural network generalizes well to unseen samples from the distribution used for training. It achieves the same accuracy within one or two order-of-magnitude fewer iterations than the original ISTA. Later works [29–36] extend this approach to different problems and network architectures and have good performance. Another recently developed technique "learning to learn" [37–39] parameterizes iterative algorithms as recurrent neural networks and shows great potential on machine learning tasks.

Applying deep unfolding on RPCA appeared recently. CORONA [23, 24] uses convolutional layers in deep unfolding and focuses on the application of ultrasound imaging. It uses SVT for low-rank approximation and mixed $\ell_{1,2}$ thresholding for outlier detection since the outliers in ultrasound imaging problems are usually structured. refRPCA [25] focuses on video foreground-background separation and also uses SVT for low-rank approximation. However, refRPCA employs a sequence of constructed reference frames to reflex correlation between consecutive frames in the video, which leads to a complicated yet learnable proximal operator for outlier detection. Given the high computational cost of SVT, both CORONA and refRPCA are not scalable. In addition, Denise [40] studies a deep unfolding RPCA method for the special case of positive semidefinite low-rank matrices. Denise achieves a remarkable speedup from baselines; however, such RPCA applications are limited.

**Contribution.** In this work, we propose a novel learning-based method, which we call Learned Robust PCA (LRPCA), for solving high-dimensional RPCA problems. Our main contributions are:

1. The proposed algorithm, LRPCA, is *scalable* and *learnable*. It uses a simple formula and differentiable operators to avoid the costly SVD and partial sorting (see Algorithm 1). LRPCA costs $\mathcal{O}(n^2 r)$ flops with a small constant[4] while all its parameters can be optimized by training.

2. An exact recovery guarantee is established for LRPCA under some mild conditions (see Theorem 1). In particular, LRPCA can tolerate outliers up to a fraction of $\alpha \lesssim \mathcal{O}(1/\mu r^{3/2}\kappa)$. Moreover, the theorem confirms that there exist a set of parameters for LRPCA to outperform the baseline algorithm ScaledGD.

3. We proposed a novel feedforward-recurrent-mixed neural network (FRMNN) model to solve RPCA. The new model first unfolds finite iterations of LRPCA and individually learns the parameters at the significant iteration; then, it learns the rules of parameter updating for subsequent iterations. Therefore, FRMNN learns the parameters for infinite iterations of LRPCA while we ensure no performance reduction from classic deep unfolding.

4. The numerical experiments confirm the advantages of LRPCA on both synthetic and real-world datasets. In particular, we successfully apply LRPCA to large video background subtraction tasks where the problem dimensions exceed the capacities of the existing learning-based RPCA approaches.

**Notation.** For any matrix $M$, $[M]_{i,j}$ denotes the $(i, j)$-th entry, $\sigma_i(M)$ denotes the $i$-th singular value, $\|M\|_1 := \sum_{i,j} |[M]_{i,j}|$ denotes the entrywise $\ell_1$ norm, $\|M\|_{\mathrm{F}} := (\sum_{i,j} [M]_{i,j}^2)^{1/2}$ denotes the Frobenius norm, $\|M\|_* := \sum_i \sigma_i(M)$ denotes the nuclear norm, $\|M\|_\infty := \max_{i,j} |[M]_{i,j}|$ denotes the largest magnitude, $\|M\|_{2,\infty} := \max_i (\sum_j [M]_{i,j}^2)^{1/2}$ denotes the largest row-wise $\ell_2$ norm, and $M^\top$ denotes the transpose. We use $\kappa := \frac{\sigma_1(X_\star)}{\sigma_r(X_\star)}$ to denote the condition number of $X_\star$.

---

[4]More precisely, LRPCA costs as low as $3n^2 r + 3n^2 + \mathcal{O}(nr^2)$ flops per iteration, which is much smaller than the hidden constant of truncated SVD.

## 2 Proposed method

In this section, we first describe the proposed learnable algorithm and then present the recovery guarantee.

We consider the non-convex minimization problem:

$$\underset{\boldsymbol{L}\in\mathbb{R}^{n_1\times r},\boldsymbol{R}\in\mathbb{R}^{n_2\times r},\boldsymbol{S}\in\mathbb{R}^{n_1\times n_2}}{\text{minimize}} \frac{1}{2}\|\boldsymbol{L}\boldsymbol{R}^\top + \boldsymbol{S} - \boldsymbol{Y}\|_{\mathrm{F}}^2, \qquad \text{subject to } \mathrm{supp}(\boldsymbol{S}) \subseteq \mathrm{supp}(\boldsymbol{S}_\star). \quad (5)$$

One may find (5) similar to the objective (3) of ScaledGD but with a different sparsity constraint for $\boldsymbol{S}$. The user may not know the true support of $\boldsymbol{S}_\star$. Also, the two constraints in (3) and (5) may seem somewhat equivalent when $\boldsymbol{S}_\star$ is assumed $\alpha$-sparse. However, we emphasize that our algorithm design does not rely on an accurate estimation of $\alpha$, and our method eliminates false-positives at every outlier-detection step.

Our algorithm proceeds in two phases: initialization and iterative updates. The first phase is done by a modified spectral initialization. In the second phase, we iteratively update outlier estimates via soft-thresholding and the factorized low-rank component estimates via scaled gradient descends. All the parameters of our algorithm are learnable during deep unfolding. The proposed algorithm, LRPCA, is summarized in Algorithm 1. We now discuss the key details of LRPCA and begin with the second phase.

---

**Algorithm 1** Learned Robust PCA (LRPCA)

---

1: **Input:** $\boldsymbol{Y} = \boldsymbol{X}_\star + \boldsymbol{S}_\star$: sparsely corrupted matrix; $r$: the rank of underlying low-rank matrix; $\{\zeta_k\}$: a set of *learned* thresholding values; $\{\eta_k\}$: a set of *learned* step sizes.
2: // Initialization:
3: $\boldsymbol{S}_0 = \mathcal{S}_{\zeta_0}(\boldsymbol{Y})$
4: $[\boldsymbol{U}_0, \boldsymbol{\Sigma}_0, \boldsymbol{V}_0] = \mathrm{SVD}_r(\boldsymbol{Y} - \boldsymbol{S}_0)$
5: $\boldsymbol{L}_0 = \boldsymbol{U}_0\boldsymbol{\Sigma}_0^{1/2}, \qquad \boldsymbol{R}_0 = \boldsymbol{V}_0\boldsymbol{\Sigma}_0^{1/2}$
6: // Iterative updates:
7: **while** Not(Stopping Condition) **do**
8:     $\boldsymbol{S}_{k+1} = \mathcal{S}_{\zeta_{k+1}}(\boldsymbol{Y} - \boldsymbol{L}_k\boldsymbol{R}_k^\top)$
9:     $\boldsymbol{L}_{k+1} = \boldsymbol{L}_k - \eta_{k+1}(\boldsymbol{L}_k\boldsymbol{R}_k^\top + \boldsymbol{S}_{k+1} - \boldsymbol{Y})\boldsymbol{R}_k(\boldsymbol{R}_k^\top\boldsymbol{R}_k)^{-1}$
10:     $\boldsymbol{R}_{k+1} = \boldsymbol{R}_k - \eta_{k+1}(\boldsymbol{L}_k\boldsymbol{R}_k^\top + \boldsymbol{S}_{k+1} - \boldsymbol{Y})^\top\boldsymbol{L}_k(\boldsymbol{L}_k^\top\boldsymbol{L}_k)^{-1}$
11: **end while**
12: **Output:** $\boldsymbol{X}_K = \boldsymbol{L}_K\boldsymbol{R}_K^\top$: the recovered low-rank matrix.

---

**Updating $\boldsymbol{S}$.** In ScaledGD [12], the sparse outlier matrix is updated by $\boldsymbol{S}_{k+1} = \mathcal{T}_{\widetilde{\alpha}}(\boldsymbol{Y} - \boldsymbol{L}_k\boldsymbol{R}_k^\top)$, where the sparsification operator $\mathcal{T}_{\widetilde{\alpha}}$ is defined in (4). Note that $\mathcal{T}_{\widetilde{\alpha}}$ requires an accurate estimate of $\alpha$ and its execution involves partial sorting on each row and column. Moreover, deep unfolding and parameter learning cannot be applied to $\mathcal{T}_{\widetilde{\alpha}}$ since it is not differentiable. Hence, we hope to find an efficient and effectively learnable operator to replace $\mathcal{T}_{\widetilde{\alpha}}$.

The well-known soft-thresholding, a.k.a. shrinkage operator,

$$[\mathcal{S}_\zeta(\boldsymbol{M})]_{i,j} = \mathrm{sign}([\boldsymbol{M}]_{i,j}) \cdot \max\{0, |[\boldsymbol{M}]_{i,j}| - \zeta\} \quad (6)$$

is our choice. Soft-thresholding has been applied as a proximal operator of $\ell_1$ norm (to a matrix entrywise) in some RPCA algorithms [2]. However, a fixed threshold, which is determined by the regularization parameter, leads to only sublinear convergence of the algorithm. Inspired by AltProj, LISTA, and their followup works [6, 10, 22, 29–34, 36], we seek for a set of thresholding parameters $\{\zeta_k\}$ that let our algorithm outperform the baseline ScaledGD.

In fact, we find that the simple soft-thresholding:

$$\boldsymbol{S}_{k+1} = \mathcal{S}_{\zeta_{k+1}}(\boldsymbol{Y} - \boldsymbol{L}_k\boldsymbol{R}_k^\top) \quad (7)$$

can also give a linear convergence guarantee if we carefully choose the thresholds, and the selected $\{\zeta_k\}$ also guarantees $\mathrm{supp}(\boldsymbol{S}_k) \subseteq \mathrm{supp}(\boldsymbol{S}_\star)$ at every iteration, i.e., no false-positive outliers. Moreover, the proposed algorithm with selected thresholds can converge even faster than ScaledGD under the very same assumptions (see Theorem 1 below for a formal statement.) Nevertheless, the theoretical selection of $\{\zeta_k\}$ relies on the knowledge of $\boldsymbol{X}_\star$, which is usually unknown to the user. Fortunately, these thresholds can be reliably learned using the deep unfolding techniques.

**Updating $X$.** To avoid the low-rank constraint on $X$, we write a rank-$r$ matrix as product of a tall and a fat matrices, that is, $X = LR^\top \in \mathbb{R}^{n_1 \times n_2}$ with $L \in \mathbb{R}^{n_1 \times r}$ and $R \in \mathbb{R}^{n_2 \times r}$. Denote the loss function $f_k := f(L_k, R_k) := \frac{1}{2}\|L_k R_k^\top + S - Y\|_{\mathrm{F}}^2$. The gradients can be easily computed:

$$\nabla_L f_k = (L_k R_k^\top + S - Y)R_k \qquad \text{and} \qquad \nabla_R f_k = (L_k R_k^\top + S - Y)^\top L_k. \qquad (8)$$

We could apply a step of gradient descent on $L$ and $R$. However, [12] finds the vanilla gradient descent approach suffers from ill-conditioning and thus introduces the scaled terms $(R_k^\top R_k)^{-1}$ and $(L_k^\top L_k)^{-1}$ to overcome this weakness. In particular, we follow ScaledGD and update the low-rank component:

$$\begin{aligned}
L_{k+1} &= L_k - \eta_{k+1}\nabla_L f_k \cdot (R_k^\top R_k)^{-1}, \\
R_{k+1} &= R_k - \eta_{k+1}\nabla_R f_k \cdot (L_k^\top L_k)^{-1},
\end{aligned} \qquad (9)$$

where $\eta_{k+1}$ is the step size at the $(k+1)$-th iteration.

**Initialization.** We use a modified spectral initialization with soft-thresholding in the proposed algorithm. That is, we first initialize the sparse matrix by $S_0 = \mathcal{S}_{\zeta_0}(Y)$, which should remove the obvious outliers. Next, for the low-rank component, we take $L_0 = U_0\Sigma_0^{1/2}$ and $R_0 = V_0\Sigma_0^{1/2}$, where $U_0\Sigma_0 V_0^\top$ is the best rank-$r$ approximation (denoted by $\mathrm{SVD}_r$) of $Y - S_0$. Clearly, the initial thresholding step is crucial for the quality of initialization and the thresholding parameter $\zeta_0$ can be optimized through learning.

For the huge-scale problems where even a single truncated SVD is computationally prohibitive, one may replace the SVD step in initialization by some batch-based low-rank approximations, e.g., CUR decomposition. While its stability lacks theoretical support when outliers appear, the empirical evidence shows that a single CUR decomposition can provide sufficiently good initialization [11].

**Computational complexity.** Along with guaranteed linear convergence, LRPCA costs only $3n^2 r + 3n^2 + \mathcal{O}(nr^2)$ flops per iteration provided $r \ll n$. In contrast, truncated SVD costs $\mathcal{O}(n^2 r)$ flops with a much larger hidden constant. The breakdown of LRPCA's complexity is provided in Appendix B.

**Limitations.** LRPCA assumes the knowledge of $\mathrm{rank}(X_\star)$, which is commonly assumed in many non-convex matrix recovery algorithms. In fact, the rank of $X_\star$ is fixed or can be easily obtained from prior knowledge in many applications, e.g., Euclidean distance matrices are rank-$(d+2)$ for the system of $d$-dimensional points [41]. However, the true rank may be hard to get in other applications, and LRPCA is not designed to learn the true rank from training data. Thus, LRPCA may have reduced performance in such applications when the target rank is seriously misestimated.

## 2.1 Theoretical results

We present the recovery guarantee of LRPCA in this subsection. Moreover, when the parameters are selected correctly, we show that LRPCA provably outperforms the state-of-the-arts.

We start with two common assumptions for RPCA:

**Assumption 1** ($\mu$-incoherence of low-rank component). *$X_\star \in \mathbb{R}^{n_1 \times n_2}$ is a rank-$r$ matrix with $\mu$-incoherence, i.e.,*

$$\|U_\star\|_{2,\infty} \le \sqrt{\frac{\mu r}{n_1}} \qquad \text{and} \qquad \|V_\star\|_{2,\infty} \le \sqrt{\frac{\mu r}{n_2}} \qquad (10)$$

*for some constant $1 \le \mu \le n$, where $U_\star \Sigma_\star V_\star^\top$ is the compact SVD of $X_\star$.*

**Assumption 2** ($\alpha$-sparsity of outlier component). *$S_\star \in \mathbb{R}^{n_1 \times n_2}$ is an $\alpha$-sparse matrix, i.e., there are at most $\alpha$ fraction of non-zero elements in each row and column of $S_\star$. In particular, we require $\alpha \lesssim \mathcal{O}(\frac{1}{\mu r^{3/2}\kappa})$ for the guaranteed recovery, which matches the requirement for ScaledGD. We future assume the problem is well-posed, i.e., $\mu, r, \kappa \ll n$.*

**Remark 1.** *Note that some applications may have a more specific structure for outliers, which may lead to a more suitable operator for $S$ updating in the particular applications. This work aims to solve the most common RPCA model with a sparsity assumption. Nevertheless, our learning framework can be adapted to another operator as long as it is differentiable.*

By the rank-sparsity uncertainty principle, a matrix cannot be incoherent and sparse simultaneously [42]. The above two assumptions ensure the uniqueness of the solution in RPCA. Now, we are ready to present our main theorem:

**Theorem 1** (Guaranteed recovery). *Suppose that $\boldsymbol{X}_\star$ is a rank-$r$ matrix with $\mu$-incoherence and $\boldsymbol{S}_\star$ is an $\alpha$-sparse matrix with $\alpha \leq \frac{1}{10^4 \mu r^{3/2} \kappa}$. If we set the thresholding values $\zeta_0 = \|\boldsymbol{X}_\star\|_\infty$ and $\zeta_k = \|\boldsymbol{L}_{k-1} \boldsymbol{R}_{k-1}^\top - \boldsymbol{X}_\star\|_\infty$ for $k \geq 1$ for LRPCA, the iterates of LRPCA satisfy*

$$\|\boldsymbol{L}_k \boldsymbol{R}_k^\top - \boldsymbol{X}_\star\|_{\mathrm{F}} \leq 0.03(1 - 0.6\eta)^k \sigma_r(\boldsymbol{X}_\star) \quad \text{and} \quad \mathrm{supp}(\boldsymbol{S}_k) \subseteq \mathrm{supp}(\boldsymbol{S}_\star), \qquad (11)$$

*with the step sizes $\eta_k = \eta \in [\frac{1}{4}, \frac{8}{9}]$.*

*Proof.* The proof of Theorem 1 is deferred to Appendix A. $\qquad\square$

Essentially, Theorem 1 states that there exists a set of selected thresholding values $\{\zeta_k\}$ that allows one to replace the sparsification operator $\mathcal{T}_{\widetilde{\alpha}}$ in ScaledGD with the simpler soft-thresholding operator $\mathcal{S}_\zeta$ and maintains the same linear convergence rate $1 - 0.6\eta$ under the same assumptions. Note that the theoretical choice of parameters relies on the knowledge of $\boldsymbol{X}_\star$—an unknown factor. Thus, Theorem 1 can be read as a proof for the existence of the appropriate parameters.

Moreover, Theorem 1 shows two advantages of LRPCA:

1. Under the very same assumptions and constants, we allow the step sizes $\eta_k$ to be as large as $\frac{8}{9}$; in contrast, ScaledGD has the step sizes no larger than $\frac{2}{3}$ [12, Theorem 2]. That is, LRPCA can provide faster convergence under the same sparsity condition, by allowing larger step sizes.

2. With the selected thresholding values, $\mathcal{S}_\zeta$ in LRPCA is effectively a projection onto $\mathrm{supp}(\boldsymbol{S}_\star)$, which matches our sparsity constrain in objective function (5). That is, $\mathcal{S}_\zeta$ takes out the larger outliers, leaves the smaller outliers, and preserves all good entries — no false-positive outlier in $\boldsymbol{S}_k$. In contrast, $\mathcal{T}_{\widetilde{\alpha}}$ necessarily yields some false-positive outliers in the earlier stages of ScaledGD, which drag the algorithm when outliers are relatively small.

**Technical innovation.** The main challenge to our analysis is to show both the distance error metric (i.e., $\mathrm{dist}(\boldsymbol{L}_k, \boldsymbol{R}_k; \boldsymbol{L}_\star, \boldsymbol{R}_\star)$, later defined in Appendix A) and $\ell_\infty$ error metric (i.e., $\|\boldsymbol{L}_k \boldsymbol{R}_k^\top - \boldsymbol{X}_\star\|_\infty$) are linearly decreasing. While the former takes some minor modifications from the proof of ScaledGD, the latter is rather challenging and utilizes several new technical lemmata. Note that ScaledGD shows $\|\boldsymbol{L}_k \boldsymbol{R}_k^\top - \boldsymbol{X}_\star\|_\infty$ is always bounded but not necessary decreasing, which is insufficient for LRPCA.

## 3 Parameter learning

Theorem 1 shows the existence of "good" parameters $\{\zeta_k\}, \{\eta_k\}$ and, in this section, we describe how to obtain such parameters via machine learning techniques.

**Feed-forward neural network.** Classic deep unfolding methods unroll an iterative algorithm and truncates it into a fixed number, says $K$, iterations. Applying such idea to our model, we regard each iteration of Algorithm 1 as a layer of a neural network and regard the variables $\boldsymbol{L}_k, \boldsymbol{R}_k, \boldsymbol{S}_k$ as the units of the $k$-th hidden layer. The top part of Figure 1 demonstrates the structure of such feed-forward neural network (FNN). The $k$-th layer is denoted as $\mathcal{L}_k$. Based on (7) and (9), it takes $\boldsymbol{Y}$ as an input and has two parameters $\zeta_k, \eta_k$ when $k \geq 1$. The initial layer $\mathcal{L}_0$ is special, it takes $\boldsymbol{Y}$ and $r$ as inputs, and it has only one parameter $\zeta_0$. For simplicity, we use $\Theta = \{\{\zeta_k\}_{k=0}^K, \{\eta_k\}_{k=1}^K\}$ to represent all parameters in this neural network.

**Training.** Given a training data set $\mathcal{D}_{\mathrm{train}}$ consisting of $(\boldsymbol{Y}, \boldsymbol{X}_\star)$ pairs (observation, ground truth), one can train the neural network and obtain parameters $\Theta$ by minimizing the following loss function:

$$\underset{\Theta}{\mathrm{minimize}} \, \mathbb{E}_{(\boldsymbol{Y}, \boldsymbol{X}_\star) \sim \mathcal{D}_{\mathrm{train}}} \|\boldsymbol{L}_K(\boldsymbol{Y}, \Theta)(\boldsymbol{R}_K(\boldsymbol{Y}, \Theta))^\top - \boldsymbol{X}_\star\|_{\mathrm{F}}^2. \qquad (12)$$

Directly minimizing (12) is called end-to-end training, and it can be easily implemented on deep learning platforms nowadays. In this paper, we adopt a more advanced training technique named as *layer-wise training* or *curriculum learning*, which has been proven as a powerful tool on training deep unfolding models [43, 44]. The process of layer-wise training is divided into $K + 1$ stages:

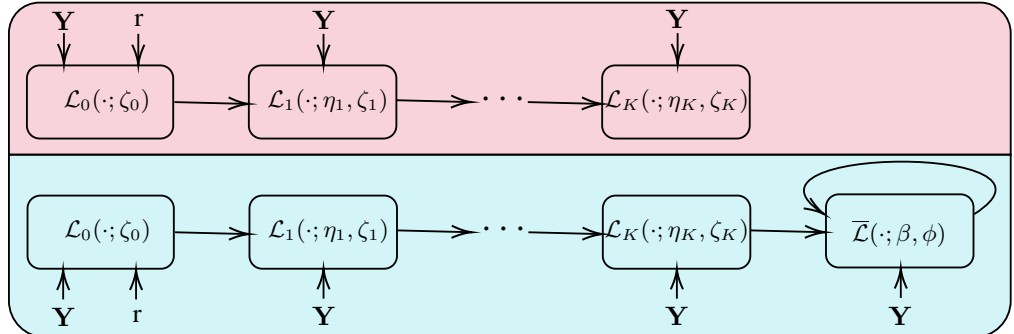

Figure 1: A high-level structure comparison between classic FNN-based deep unfolding (top) and proposed FRMNN-based deep unfolding (bottom). In the diagrams, $\mathcal{L}_k$ denotes the $k$-th layer of FNN and $\overline{\mathcal{L}}$ is a layer of RNN.

- Training the 0-th layer with $\text{minimize}_\Theta \, \mathbb{E}\|\boldsymbol{L}_0 \boldsymbol{R}_0^\top - \boldsymbol{X}_\star\|_{\mathrm{F}}^2$.
- Training the 1-st layer with $\text{minimize}_\Theta \, \mathbb{E}\|\boldsymbol{L}_1 \boldsymbol{R}_1^\top - \boldsymbol{X}_\star\|_{\mathrm{F}}^2$.
- $\cdots$
- Training the final layer with (12): $\text{minimize}_\Theta \, \mathbb{E}\|\boldsymbol{L}_K \boldsymbol{R}_K^\top - \boldsymbol{X}_\star\|_{\mathrm{F}}^2$.

**Feedforward-recurrent-mixed neural network.** One disadvantage of the aforementioned FNN model is its fixed number of layers. If one wants to go further steps and obtain higher accuracy after training a neural network, one may have to retrain the neural network once more. Recurrent neural network (RNN) has tied parameters over different layers and, consequently, is extendable to infinite layers. However, we find that the starting iterations play significant roles in the convergence (validated in Section 4 later) and their parameters should be trained individually. Thus, we propose a hybrid model that is demonstrated in the bottom part of Figure 1, named as Feedforward-recurrent-mixed neural network (FRMNN). We use a recurrent neural network appended after a $K$-layer feedforward neural network. When $k \geq K$, in the $(k-K)$-th loop of the RNN layer, we follow the same calculation procedures with Algorithm 1 and determine the parameters $\zeta_k$ and $\eta_k$ by

$$\eta_k = \beta\eta_{k-1} \quad \text{and} \quad \zeta_k = \phi\zeta_{k-1}. \tag{13}$$

Thus, all the RNN layers share the common parameters $\beta$ and $\phi$.

The training of FRMNN follows in two phases:

- Training the $K$-layer FNN with layer-wise training.
- Fixing the FNN and searching RNN parameters $\beta$ and $\phi$ to minimize the convergence metric at the $(\overline{K} - K)$-th layer of RNN for some $\overline{K} > K$:

$$\underset{\beta,\phi}{\text{minimize}} \, \mathbb{E}_{(\boldsymbol{Y}, \boldsymbol{X}_\star) \sim \mathcal{D}_{\text{train}}} \|\boldsymbol{L}_{\overline{K}}(\beta, \phi)(\boldsymbol{R}_{\overline{K}}(\beta, \phi))^\top - \boldsymbol{X}_\star\|_{\mathrm{F}}^2. \tag{14}$$

Our new model provides the flexibility of training a neural network with finite $\overline{K}$ layers and testing it with infinite layers. Consequently, the stop condition of LRPCA has to be (Run $K$ iterations) if the parameters are trained via FNN model; and the stop condition can be (Error < Tolerance) if our FRMNN model is used.

In this paper, we use stochastic gradient descent (SGD) in the layer-wise training stage and use grid search to obtain $\beta$ and $\phi$ since there are only two parameters. Moreover, we find that picking $K + 3 \leq \overline{K} \leq K + 5$ is empirically good.

## 4 Numerical experiments

In this section, we compare the empirical performance of LRPCA with the state-of-the-arts: ScaledGD [12] and AltProj [6]. We hand tune the parameters for ScaledGD and AltProj to achieve their best

performance in the experiments. Moreover, all speed tests are executed on a Windows 10 laptop with Intel i7-8750H CPU, 32GB RAM and Nvidia GTX-1060 GPU. The parameters learning processes are executed on an Ubuntu workstation with Intel i9-9940X CPU, 128GB RAM and two Nvidia RTX-3080 GPUs. All our synthetic test results are averaged over 50 random generated instances, and details of random instance generation can be found in Appendix C. The code for LRPCA is available online at `https://github.com/caesarcai/LRPCA`.

**Unfolding models.** We compare the performance of LRPCA with the parameters learned from different unfolding models: classic FNN, RNN and proposed FRMNN. In particular, FNN model unrolls 10 iterations of LRPCA and RNN model directly starts the training on the second phase of FRMNN, i.e., $K = 0$. For FRMNN model, we take $K = 10$ and $\overline{K} = 15$ for the training. The test results is summarized in Figure 2. One can see FRMNN model extends FNN model to infinite layers with performance reduction while

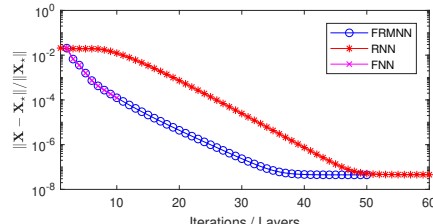

Figure 2: Convergence comparison for FNN-based, RNN-based, and FRMNN-based learning.

the pure RNN model drops the convergence performance. Note that the convergence curves of both FRMNN and RNN go down till they reach $\mathcal{O}(10^{-8})$ which is the machine precision since we use single precision in this experiment.

**Computational efficiency.** In this section, we present the speed advantage of LRPCA. First, we compare the convergence behavior of LRPCA to our baseline ScaledGD in Figure 3. We find LRPCA converges in much less iterations, especially when the outlier sparsity $\alpha$ is larger. Next, we compare per-iteration runtime of LRPCA to our baseline ScaledGD in Figure 4. One can see that ScaledGD runs slower when $\alpha$ becomes larger. In contrast, the per-iteration runtime of LRPCA is insensitive to $\alpha$ and is significantly faster than ScaledGD. Finally, we compare the total runtime among LRPCA, ScaledGD and AltProj in Figure 5. Therein, we find LRPCA is substantially faster than the state-of-the-arts if the model has been trained. The training time for models with different sizes and settings are reported in the supplement.

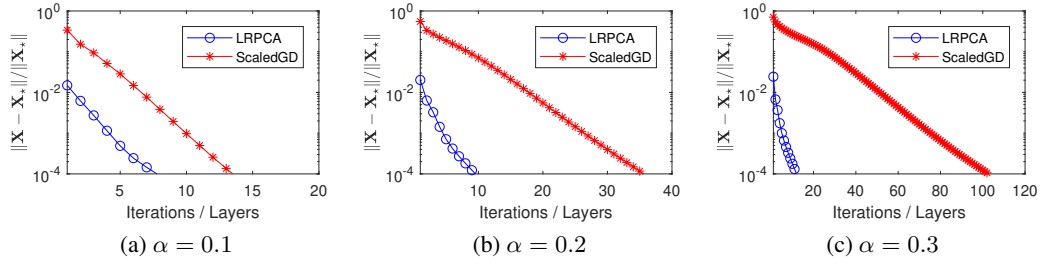

(a) $\alpha = 0.1$      (b) $\alpha = 0.2$      (c) $\alpha = 0.3$

Figure 3: Convergence comparison for LRPCA and baseline ScaledGD with varying outlier sparsity $\alpha$. Problem dimension $n = 1000$ and rank $r = 5$.

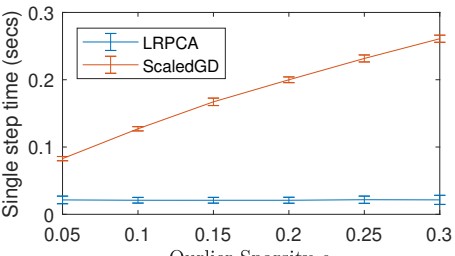
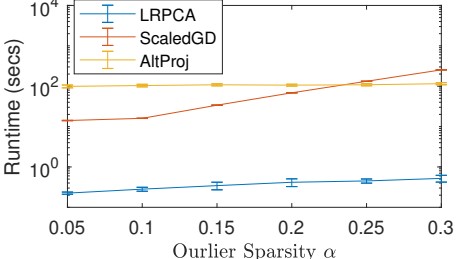

Figure 4: Single-step runtime comparison with error bar for LRPCA and baseline ScaledGD with varying outlier sparsity $\alpha$. Problem dimension $n = 1000$ and rank $r = 5$. The results are based on running 100 iterations excluding initialization.

Figure 5: Runtime comparison with error bar for LRPCA and start-of-the-arts (ScaledGD and AltProj) with varying outlier sparsity $\alpha$. Problem dimension $d = 3000$ and rank $r = 5$. All algorithms halt when $\|\boldsymbol{Y} - \boldsymbol{X} - \boldsymbol{S}\|_{\mathrm{F}}/\|\boldsymbol{Y}\|_{\mathrm{F}} < 10^{-4}$.

**Recovery performance.** We generate 10 problems for each of the outlier density levels (i.e., $\alpha$) and compare the recoverabilities of LRPCA and ScaledGD. The test result is presented in the Table 1, where we can conclude that LRPCA is more robust when $\alpha$ is larger.

Table 1: Recovery performance under different outlier density levels.

| $\alpha$ | 0.4 | 0.45 | 0.5 | 0.55 | 0.6 | 0.65 | 0.7 |
|---|---|---|---|---|---|---|---|
| LRPCA | **10**/10 | **10**/10 | **10**/10 | **9**/10 | **8**/10 | 0/10 | 0/10 |
| ScaledGD | **10**/10 | **10**/10 | 0/10 | 0/10 | 0/10 | 0/10 | 0/10 |

**Video background subtraction.** In this section, we apply LRPCA to the task of video background subtraction. We use VIRAT video dataset[5] as our benchmark, which includes numerous colorful videos with static backgrounds. The videos have been grouped into training and testing sets. We used 65 videos for parameter learning and 4 videos for verification[6]. We first convert all videos to grayscale. Next, each frame of the videos is vectorized and become a matrix column, and all frames of a video form a data matrix. The static backgrounds are the low-rank component of the data matrices and moving objects can be viewed as outliers, thus we can separate the backgrounds and foregrounds via RPCA. The video details and the experimental results are summarized in Table 2 and selected visual results for LRPCA are presented in Figure 6. One can see LRPCA runs much faster than both ScaledGD and AltProj in all verification tests. Furthermore, the background subtraction results of LRPCA are also visually desirable. More details about the experimental setup can be found in Appendix C, alone with additional visual results for ScaledGD and AltProj.

Table 2: Video details and runtime comparison for the task of background subtraction. All algorithms halt when $\max(\|\boldsymbol{X}_k - \boldsymbol{X}_{k-1}\|_{\mathrm{F}}/\|\boldsymbol{X}_{k-1}\|_{\mathrm{F}}, \|\boldsymbol{S}_k - \boldsymbol{S}_{k-1}\|_{\mathrm{F}}/\|\boldsymbol{S}_{k-1}\|_{\mathrm{F}}) < 10^{-3}$.

| VIDEO NAME | FRAME SIZE | FRAME NUMBER | RUNTIME (secs) | | |
|---|---|---|---|---|---|
| | | | LRPCA | ScaledGD | AltProj |
| ParkingLot1 | $320 \times 180$ | 965 | **16.01** | 260.45 | 63.04 |
| ParkingLot2 | $320 \times 180$ | 2149 | **33.95** | 639.03 | 144.50 |
| ParkingLot3 | $480 \times 270$ | 1110 | **38.85** | 662.08 | 166.91 |
| StreeView | $480 \times 270$ | 1034 | **33.73** | 626.05 | 167.66 |

**Ultrasound imaging.** We compare LRPCA with CORONA [23], a state-of-the-art learning-based RPCA method, on ultrasound imaging. The testing dataset consists of 2400 training examples and 800 validation examples, which can be downloaded online at `https://www.wisdom.weizmann.ac.il/~yonina`. Each example is of size $1024 \times 40$. The target rank of the low-rank matrix is set to be $r = 1$. Recovery accuracy is measured by the loss function: $\mathrm{loss} = \mathrm{MSE}(X_{\mathrm{output}}, X_\star)$, where MSE stands for the mean square error. The test results are summarized in Table 3.

Table 3: Test results for ultrasound imaging.

| ALGORITHM | AVERAGE INFERENCE TIME | AVERAGE loss |
|---|---|---|
| LRPCA | **0.0057** secs | $9.97 \times 10^{-4}$ |
| CORONA | 0.9225 secs | **4.88** $\times 10^{-4}$ |

Note that LRPCA is designed for generic RPCA and CORONA is specifically designed for ultrasound imaging. Thus, it is not a surprise that our recovery accuracy is slightly worse than CORONA.

---

[5] Available at: `https://viratdata.org`.

[6] The names of the tested videos are corresponding to these original video numbers in VIRAT dataset: ParkingLot1: S_000204_06_001527_001560, ParkingLot2: S_010100_05_000917_001017, ParkingLot3: S_050204_00_000000_000066, StreeView: S_050100_13_002202_002244.

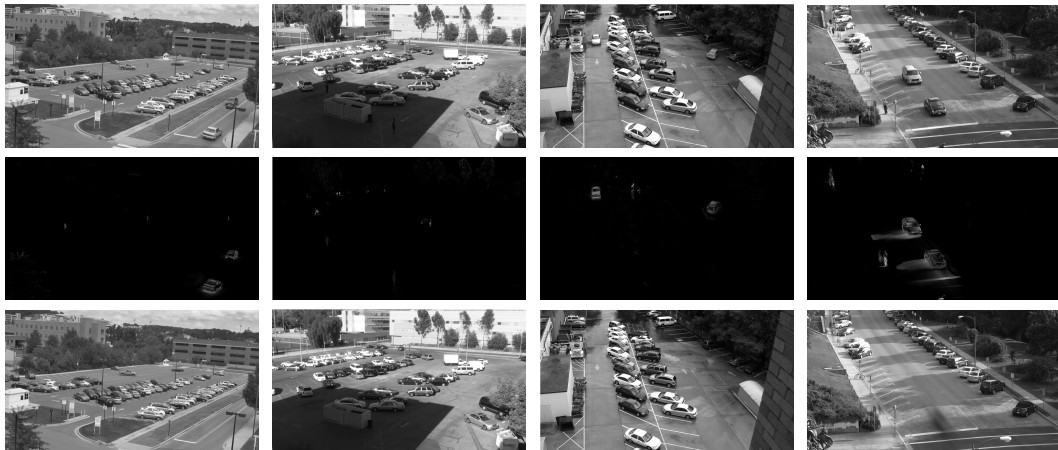

Figure 6: Video background subtraction visual results for LRPCA. Each column represents a selected frame from the tested videos. From left to right, they are ParkingLot1, ParkingLot2, ParkingLot3, and StreetView. The first row contains the original frames. The next two rows are the separated foreground and background produced by LRPCA, respectively. More visual results for ScaledGD and AltProj are available in Appendix C.

However, the average runtime of LRPCA is substantially faster than CORONA. We believe that our speed advantage will be even more significant on larger-scale examples, which is indeed our main contribution: scalability.

**More numerical experiments.** There are some other interesting empirical problems worth to explore and we put the extra experimental results in Appendix C due to the space limitation. They are summarized here: 1. Our model has good generalization ability. We can train our model on small-size and low-rank training instances (say, $n = 1000$ and $r = 5$) and use the model DIRECTLY on problems with larger size (say, $n = 3000$ and $r = 5$) or with higher rank (say, $n = 1000$ and $r = 15$) with good performance. 2. The learned parameters are explainable. With the learned parameters, LRPCA goes very aggressively with large step sizes in the first several steps and tends to be conservative after that.

## 5 Conclusion and future work

We have presented a highly efficient learning-based approach for high-dimensional RPCA problems. In addition, we have presented a novel feedforward-recurrent-mixed neural network model which can learn the parameters for potentially infinite iterations without performance reduction. Theoretical recovery guarantee has been established for the proposed algorithm. The numerical experiments show that the proposed approach is superior to the state-of-the-arts.

One of our future directions is to study learning-based stochastic approach. While this paper focuses on deterministic RPCA approaches, the stochastic RPCA approaches, e.g., partialGD [8], PG-RMC [9], and IRCUR [11], have shown promising speed advantage, especially for large-scale problems. Another future direction is to explore robust tensor decomposition incorporate with deep learning, as some preliminary studies have shown the advantages of tensor structure in certain machine learning tasks [45, 46].

## Broader impact

RPCA has been broadly applied as one of fundamental techniques in data mining and machine learning communities. Hence, through its applications, the proposed learning-based approach for RPCA has potential broader impacts that data mining and machine learning have, as well as their limitations.

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
