# Supplementary Material for
# Learned Robust PCA: A Scalable Deep Unfolding Approach for High-Dimensional Outlier Detection

## A   Proofs

In this section, we provide the mathematical proofs for the claimed theoretical results. Note that the proof of our convergence theorem follows the route established in [12]. However, the details of our proof are quite involved since we replaced the sparsification operator which substantially changes the method of outlier detection.

Let $\boldsymbol{L}_\star := \boldsymbol{U}_\star \boldsymbol{\Sigma}_\star^{1/2}$ and $\boldsymbol{R}_\star := \boldsymbol{V}_\star \boldsymbol{\Sigma}_\star^{1/2}$ where $\boldsymbol{U}_\star \boldsymbol{\Sigma}_\star \boldsymbol{V}_\star^\top$ is the compact SVD of $\boldsymbol{X}_\star$. For theoretical analysis, we consider the error metric for decomposed rank-$r$ matrices:

$$\text{dist}(\boldsymbol{L}, \boldsymbol{R}; \boldsymbol{L}_\star, \boldsymbol{R}_\star) := \inf_{\boldsymbol{Q} \in \mathbb{R}^{r \times r}, \text{rank}(\boldsymbol{Q})=r} \left( \|(\boldsymbol{L}\boldsymbol{Q} - \boldsymbol{L}_\star)\boldsymbol{\Sigma}_\star^{1/2}\|_F^2 + \|(\boldsymbol{R}\boldsymbol{Q}^{-\top} - \boldsymbol{R}_\star)\boldsymbol{\Sigma}_\star^{1/2}\|_F^2 \right)^{1/2}.$$

Notice that the optimal alignment matrix $\boldsymbol{Q}$ exists and invertible if $\boldsymbol{L}$ and $\boldsymbol{R}$ are sufficiently close to $\boldsymbol{L}_\star$ and $\boldsymbol{R}_\star$. In particular, one can have the following lemma.

**Lemma 2** ([12, Lemma 9]). *For any $\boldsymbol{L} \in \mathbb{R}^{n_1 \times r}$ and $\boldsymbol{R} \in \mathbb{R}^{n_2 \times r}$, if*

$$\text{dist}(\boldsymbol{L}, \boldsymbol{R}; \boldsymbol{L}_\star, \boldsymbol{R}_\star) < c\sigma_r(\boldsymbol{X}_\star)$$

*for some $0 < c < 1$, then the optimal alignment matrix $\boldsymbol{Q}$ between $[\boldsymbol{L}, \boldsymbol{R}]$ and $[\boldsymbol{L}_\star, \boldsymbol{R}_\star]$ exists and is invertible.*

**More notation.**   In addition to the notation introduced in Section 1, we provide some more notation for the analysis. $\vee$ denotes the logical disjunction, which takes the max of two operands. For any matrix $\boldsymbol{M}$, $\|\boldsymbol{M}\|_2 = \sigma_1(\boldsymbol{M})$ denotes the spectral norm and $\|\boldsymbol{M}\|_{1,\infty} = \max_i \sum_j |[\boldsymbol{M}]_{i,j}|$ denotes the largest row-wise $\ell_1$ norm.

For ease of presentation, we take $n := n_1 = n_2$ in the rest of this section, but we emphasize that similar results can be easily drawn for the rectangular matrix setting. Furthermore, we introduce following shorthand for notational convenience: $\boldsymbol{Q}_k$ denotes the optimal alignment matrix between $(\boldsymbol{L}_k, \boldsymbol{R}_k)$ and $(\boldsymbol{L}_\star, \boldsymbol{R}_\star)$, $\boldsymbol{L}_\natural := \boldsymbol{L}_k \boldsymbol{Q}_k$, $\boldsymbol{R}_\natural := \boldsymbol{R}_k \boldsymbol{Q}_k^{-\top}$, $\boldsymbol{\Delta}_L := \boldsymbol{L}_\natural - \boldsymbol{L}_\star$, $\boldsymbol{\Delta}_R := \boldsymbol{R}_\natural - \boldsymbol{R}_\star$, and $\boldsymbol{\Delta}_S := \boldsymbol{S}_{k+1} - \boldsymbol{S}_\star$.

### A.1   Proof of Theorem 1

We first present the theorems of local linear convergence and guaranteed initialization. The proofs of these two theorems can be find in Sections A.3 and A.4, respectively.

**Theorem 3** (Local linear convergence). *Suppose that $\boldsymbol{X}_\star = \boldsymbol{L}_\star \boldsymbol{R}_\star^\top$ is a rank-$r$ matrix with $\mu$-incoherence and $\boldsymbol{S}_\star$ is an $\alpha$-sparse matrix with $\alpha \leq \frac{1}{10^4 \mu r^{1.5}}$. Let $\boldsymbol{Q}_k$ be the optimal alignment matrix between $[\boldsymbol{L}_k, \boldsymbol{R}_k]$ and $[\boldsymbol{L}_\star, \boldsymbol{R}_\star]$. If the initial guesses obey the conditions*

$$\text{dist}(\boldsymbol{L}_0, \boldsymbol{R}_0; \boldsymbol{L}_\star, \boldsymbol{R}_\star) \leq \varepsilon_0 \sigma_r(\boldsymbol{X}_\star),$$

$$\|(\boldsymbol{L}_0 \boldsymbol{Q}_0 - \boldsymbol{L}_\star)\boldsymbol{\Sigma}_\star^{1/2}\|_{2,\infty} \vee \|(\boldsymbol{R}_0 \boldsymbol{Q}_0^{-\top} - \boldsymbol{R}_\star)\boldsymbol{\Sigma}_\star^{1/2}\|_{2,\infty} \leq \sqrt{\frac{\mu r}{n}} \sigma_r(\boldsymbol{X}_\star)$$

*with $\varepsilon_0 := 0.02$, then by setting the thresholding values $\zeta_k = \|\boldsymbol{X}_\star - \boldsymbol{L}_{k-1}\boldsymbol{R}_{k-1}^\top\|_\infty$ and the fixed step size $\eta_k = \eta \in [\frac{1}{4}, \frac{8}{9}]$, the iterates of Algorithm 1 satisfy*

$$\text{dist}(\boldsymbol{L}_k, \boldsymbol{R}_k; \boldsymbol{L}_\star, \boldsymbol{R}_\star) \leq \varepsilon_0 \tau^k \sigma_r(\boldsymbol{X}_\star),$$

$$\|(\boldsymbol{L}_k \boldsymbol{Q}_k - \boldsymbol{L}_\star)\boldsymbol{\Sigma}_\star^{1/2}\|_{2,\infty} \vee \|(\boldsymbol{R}_k \boldsymbol{Q}_k^{-\top} - \boldsymbol{R}_\star)\boldsymbol{\Sigma}_\star^{1/2}\|_{2,\infty} \leq \sqrt{\frac{\mu r}{n}} \tau^k \sigma_r(\boldsymbol{X}_\star),$$

*where the convergence rate $\tau := 1 - 0.6\eta$.*

**Theorem 4** (Guaranteed initialization). *Suppose that $X_\star = L_\star R_\star^\top$ is a rank-$r$ matrix with $\mu$-incoherence and $S_\star$ is an $\alpha$-sparse matrix with $\alpha \leq \frac{c_0}{\mu r^{1.5}\kappa}$ for some small positive constant $c_0 \leq \frac{1}{35}$. Let $Q_0$ be the optimal alignment matrix between $[L_0, R_0]$ and $[L_\star, R_\star]$. By setting the thresholding values $\zeta_0 = \|X_\star\|_\infty$, the initial guesses satisfy*

$$\mathrm{dist}(L_0, R_0; L_\star, R_\star) \leq 10c_0\sigma_r(X_\star),$$

$$\|(L_0 Q_0 - L_\star)\Sigma_\star^{1/2}\|_{2,\infty} \vee \|(R_0 Q_0^{-\top} - R_\star)\Sigma_\star^{1/2}\|_{2,\infty} \leq \sqrt{\frac{\mu r}{n}}\sigma_r(X_\star).$$

In addition, we present a lemma that verifies our selection of thresholding values is indeed effective.

**Lemma 5.** *At the $(k+1)$-th iteration of Algorithm 1, taking the thresholding value $\zeta_{k+1} := \|X_\star - X_k\|_\infty$ gives*

$$\|S_\star - S_{k+1}\|_\infty \leq 2\|X_\star - X_k\|_\infty \qquad \text{and} \qquad \mathrm{supp}(S_{k+1}) \subseteq \mathrm{supp}(S_\star).$$

*Proof.* Denote $\Omega_\star := \mathrm{supp}(S_\star)$ and $\Omega_{k+1} := \mathrm{supp}(S_{k+1})$. Recall that $S_{k+1} = \mathcal{S}_{\zeta_{k+1}}(Y - X_k) = \mathcal{S}_{\zeta_{k+1}}(S_\star + X_\star - X_k)$. Since $[S_\star]_{i,j} = 0$ outside its support, so $[Y - X_k]_{i,j} = [X_\star - X_k]_{i,j}$ for the entries $(i, j) \in \Omega_\star^c$. Applying the chosen thresholding value $\zeta_{k+1} := \|X_\star - X_k\|_\infty$, one have $[S_{k+1}]_{i,j} = 0$ for all $(i, j) \in \Omega_\star^c$. Hence, the support of $S_{k+1}$ must belongs to the support of $S_\star$, i.e.,

$$\mathrm{supp}(S_{k+1}) = \Omega_{k+1} \subseteq \Omega_\star = \mathrm{supp}(S_\star).$$

This proves our first claim.

Obviously, $[S_\star - S_{k+1}]_{i,j} = 0$ for all $(i, j) \in \Omega_\star^c$. Moreover, we can split the entries in $\Omega_\star$ into two groups:

$$\Omega_{k+1} = \{(i, j) \mid |[Y - X_k]_{i,j}| > \zeta_{k+1} \text{ and } [S_\star]_{i,j} \neq 0\} \quad \text{and}$$
$$\Omega_\star \backslash \Omega_{k+1} = \{(i, j) \mid |[Y - X_k]_{i,j}| \leq \zeta_{k+1} \text{ and } [S_\star]_{i,j} \neq 0\},$$

and it holds

$$|[S_\star - S_{k+1}]_{i,j}| = \begin{cases} |[X_k - X_\star]_{i,j} - \mathrm{sign}([Y - X_k]_{i,j})\zeta_{k+1}| \\ |[S_\star]_{i,j}| \end{cases}$$
$$\leq \begin{cases} |[X_k - X_\star]_{i,j}| + \zeta_{k+1} \\ |[X_\star - X_k]_{i,j}| + \zeta_{k+1} \end{cases}$$
$$\leq \begin{cases} 2\|X_\star - X_k\|_\infty & (i, j) \in \Omega_{k+1}, \\ 2\|X_\star - X_k\|_\infty & (i, j) \in \Omega_\star \backslash \Omega_{k+1}. \end{cases}$$

Therefore, we can conclude $\|S_\star - S_{k+1}\|_\infty \leq 2\|X_\star - X_k\|_\infty$. $\qquad\square$

Now, we are already to prove Theorem 1.

*Proof of Theorem 1.* Take $c_0 = 10^{-4}$ in Theorem 4. Thus, the results of Theorem 4 satisfy the condition of Theorem 3, and gives

$$\mathrm{dist}(L_k, R_k; L_\star, R_\star) \leq 0.02(1 - 0.6\eta)^k \sigma_r(X_\star),$$

$$\|(L_k Q_k - L_\star)\Sigma_\star^{1/2}\|_{2,\infty} \vee \|(R_k Q_k^{-\top} - R_\star)\Sigma_\star^{1/2}\|_{2,\infty} \leq \sqrt{\frac{\mu r}{n}}(1 - 0.6\eta)^k \sigma_r(X_\star)$$

for all $k \geq 0$. [12, Lemma 3] states that

$$\|L_k R_k^\top - X_\star\|_F \leq 1.5\,\mathrm{dist}(L_k, R_k; L_\star, R_\star)$$

as long as $\|(L_k Q_k - L_\star)\Sigma_\star^{1/2}\|_{2,\infty} \vee \|(R_k Q_k^{-\top} - R_\star)\Sigma_\star^{1/2}\|_{2,\infty} \leq \sqrt{\frac{\mu r}{n}}\sigma_r(X_\star)$. Hence, our first claim is proved.

When $k \geq 1$, the second claim is directly followed by Lemma 5. When $k = 0$, take $X_{-1} = 0$, then one can see $S_0 = \mathcal{S}_{\zeta_0}(Y) = \mathcal{S}_{\zeta_0}(Y - X_{-1})$ where $\zeta_0 = \|X_\star\|_\infty = \|X_\star - X_{-1}\|_\infty$. Applying Lemma 5 again, we have the second claim for all $k \geq 0$.

This finishes the proof. $\qquad\square$

## A.2 Auxiliary lemmata

Before we can present the proofs for Theorems 3 and 4, several important auxiliary lemmata must be processed.

**Lemma 6.** *For any $\alpha$-sparse matrix $\boldsymbol{S} \in \mathbb{R}^{n \times n}$, the following inequalities hold:*

$$\|\boldsymbol{S}\|_2 \leq \alpha n \|\boldsymbol{S}\|_\infty,$$
$$\|\boldsymbol{S}\|_{2,\infty} \leq \sqrt{\alpha n} \|\boldsymbol{S}\|_\infty,$$
$$\|\boldsymbol{S}\|_{1,\infty} \leq \alpha n \|\boldsymbol{S}\|_\infty.$$

*Proof.* The first claim has been shown as [6, Lemma 4]. The rest two claims are directly followed by the fact $\boldsymbol{S}$ has at most $\alpha n$ non-zero elements in each row and each column. $\square$

**Lemma 7.** *If*

$$\mathrm{dist}(\boldsymbol{L}_k, \boldsymbol{R}_k; \boldsymbol{L}_\star, \boldsymbol{R}_\star) \leq \varepsilon_0 \tau^k \sigma_r(\boldsymbol{X}_\star),$$

*then the following inequalities hold*

$$\|\boldsymbol{\Delta}_L \boldsymbol{\Sigma}_\star^{1/2}\|_{\mathrm{F}} \vee \|\boldsymbol{\Delta}_R \boldsymbol{\Sigma}_\star^{1/2}\|_{\mathrm{F}} \leq \varepsilon_0 \tau^k \sigma_r(\boldsymbol{X}_\star)$$
$$\|\boldsymbol{\Delta}_L \boldsymbol{\Sigma}_\star^{1/2}\|_2 \vee \|\boldsymbol{\Delta}_R \boldsymbol{\Sigma}_\star^{1/2}\|_2 \leq \varepsilon_0 \tau^k \sigma_r(\boldsymbol{X}_\star)$$

*Proof.* Recall that $\mathrm{dist}(\boldsymbol{L}_k, \boldsymbol{R}_k; \boldsymbol{L}_\star, \boldsymbol{R}_\star) = \sqrt{\|\boldsymbol{\Delta}_L \boldsymbol{\Sigma}_\star^{1/2}\|_{\mathrm{F}}^2 \vee \|\boldsymbol{\Delta}_R \boldsymbol{\Sigma}_\star^{1/2}\|_{\mathrm{F}}^2}$. The first claim is directly followed by the definition of $\mathrm{dist}$.

By the fact that $\|\boldsymbol{A}\|_2 \leq \|\boldsymbol{A}\|_{\mathrm{F}}$ for any matrix, we deduct the second claim from the first claim. $\square$

**Lemma 8.** *If*

$$\mathrm{dist}(\boldsymbol{L}_k, \boldsymbol{R}_k; \boldsymbol{L}_\star, \boldsymbol{R}_\star) \leq \varepsilon_0 \tau^k \sigma_r(\boldsymbol{X}_\star),$$

*then it holds*

$$\|\boldsymbol{L}_\natural(\boldsymbol{L}_\natural^\top \boldsymbol{L}_\natural)^{-1} \boldsymbol{\Sigma}_\star^{1/2}\|_2 \vee \|\boldsymbol{R}_\natural(\boldsymbol{R}_\natural^\top \boldsymbol{R}_\natural)^{-1} \boldsymbol{\Sigma}_\star^{1/2}\|_2 \leq \frac{1}{1 - \varepsilon_0}.$$

*Proof.* [12, Lemma 12] provides the following inequalities:

$$\|\boldsymbol{L}_\natural(\boldsymbol{L}_\natural^\top \boldsymbol{L}_\natural)^{-1} \boldsymbol{\Sigma}_\star^{1/2}\|_2 \leq \frac{1}{1 - \|\boldsymbol{\Delta}_L \boldsymbol{\Sigma}_\star^{-1/2}\|_2},$$
$$\|\boldsymbol{R}_\natural(\boldsymbol{R}_\natural^\top \boldsymbol{R}_\natural)^{-1} \boldsymbol{\Sigma}_\star^{1/2}\|_2 \leq \frac{1}{1 - \|\boldsymbol{\Delta}_R \boldsymbol{\Sigma}_\star^{-1/2}\|_2},$$

as long as $\|\boldsymbol{\Delta}_L \boldsymbol{\Sigma}_\star^{-1/2}\|_2 \vee \|\boldsymbol{\Delta}_R \boldsymbol{\Sigma}_\star^{-1/2}\|_2 < 1$.

By Lemma 7, we have $\|\boldsymbol{\Delta}_L \boldsymbol{\Sigma}_\star^{-1/2}\|_2 \vee \|\boldsymbol{\Delta}_R \boldsymbol{\Sigma}_\star^{-1/2}\|_2 \leq \varepsilon_0 \tau^k \leq \varepsilon_0$, given $\tau = 1 - 0.6\eta < 1$. The proof is finished since $\varepsilon_0 = 0.02 < 1$. $\square$

**Lemma 9.** *If*

$$\|(\boldsymbol{L}_{k+1} \boldsymbol{Q}_k - \boldsymbol{L}_\star) \boldsymbol{\Sigma}_\star^{1/2}\|_2 \vee \|(\boldsymbol{R}_{k+1} \boldsymbol{Q}_k^{-\top} - \boldsymbol{R}_\star) \boldsymbol{\Sigma}_\star^{1/2}\|_2 \leq \varepsilon_0 \tau^{k+1} \sigma_r(\boldsymbol{X}_\star),$$

*then*

$$\|\boldsymbol{\Sigma}_\star^{1/2} \boldsymbol{Q}_k^{-1} (\boldsymbol{Q}_{k+1} - \boldsymbol{Q}_k) \boldsymbol{\Sigma}_\star^{1/2}\|_2 \vee \|\boldsymbol{\Sigma}_\star^{1/2} \boldsymbol{Q}_k^\top (\boldsymbol{Q}_{k+1} - \boldsymbol{Q}_k)^{-\top} \boldsymbol{\Sigma}_\star^{1/2}\|_2 \leq \frac{2\varepsilon_0}{1 - \varepsilon_0} \sigma_r(\boldsymbol{X}_\star).$$

*Proof.* [12, Lemma 14] provides the inequalities:

$$\|\boldsymbol{\Sigma}_\star^{1/2} \widetilde{\boldsymbol{Q}}^{-1} \widehat{\boldsymbol{Q}} \boldsymbol{\Sigma}_\star^{1/2} - \boldsymbol{\Sigma}_\star\|_2 \leq \frac{\|\boldsymbol{R}(\widetilde{\boldsymbol{Q}}^{-\top} - \widehat{\boldsymbol{Q}}^{-\top}) \boldsymbol{\Sigma}_\star^{1/2}\|_2}{1 - \|(\boldsymbol{R}\widehat{\boldsymbol{Q}}^{-\top} - \boldsymbol{R}_\star) \boldsymbol{\Sigma}_\star^{-1/2}\|_2}$$

$$\|\mathbf{\Sigma}_\star^{1/2}\widetilde{\mathbf{Q}}^\top\widehat{\mathbf{Q}}^{-\top}\mathbf{\Sigma}_\star^{1/2} - \mathbf{\Sigma}_\star\|_2 \le \frac{\|\mathbf{L}(\widetilde{\mathbf{Q}} - \widehat{\mathbf{Q}})\mathbf{\Sigma}_\star^{1/2}\|_2}{1 - \|(\mathbf{L}\widehat{\mathbf{Q}} - \mathbf{L}_\star)\mathbf{\Sigma}_\star^{-1/2}\|_2}$$

for any $\mathbf{L}, \mathbf{R} \in \mathbb{R}^{n\times r}$ and invertible $\widetilde{\mathbf{Q}}, \widehat{\mathbf{Q}} \in \mathbb{R}^{r\times r}$, as long as $\|(\mathbf{L}\widehat{\mathbf{Q}} - \mathbf{L}_\star)\mathbf{\Sigma}_\star^{-1/2}\|_2 \vee \|(\mathbf{R}\widehat{\mathbf{Q}}^{-\top} - \mathbf{L}_\star)\mathbf{\Sigma}_\star^{-1/2}\|_2 < 1$.

We will focus on the first term for now. By the assumption of this lemma and the definition of $\mathbf{Q}_{k+1}$, we have

$$\|(\mathbf{R}_{k+1}\mathbf{Q}_k^{-\top} - \mathbf{R}_\star)\mathbf{\Sigma}_\star^{1/2}\|_2 \le \varepsilon_0\tau^{k+1}\sigma_r(\mathbf{X}_\star),$$
$$\|(\mathbf{R}_{k+1}\mathbf{Q}_{k+1}^{-\top} - \mathbf{R}_\star)\mathbf{\Sigma}_\star^{1/2}\|_2 \le \varepsilon_0\tau^{k+1}\sigma_r(\mathbf{X}_\star),$$
$$\|(\mathbf{R}_{k+1}\mathbf{Q}_{k+1}^{-\top} - \mathbf{R}_\star)\mathbf{\Sigma}_\star^{-1/2}\|_2 \le \varepsilon_0\tau^{k+1}.$$

Thus, by taking $\mathbf{R} = \mathbf{R}_{k+1}$, $\widetilde{\mathbf{Q}} = \mathbf{Q}_k$, and $\widehat{\mathbf{Q}} = \mathbf{Q}_{k+1}$, we obtain

$$\begin{aligned}
\|\mathbf{\Sigma}_\star^{1/2}\mathbf{Q}_k^{-1}(\mathbf{Q}_{k+1} - \mathbf{Q}_k)\mathbf{\Sigma}_\star^{1/2}\|_2 &= \|\mathbf{\Sigma}_\star^{1/2}\mathbf{Q}_k^{-1}\mathbf{Q}_{k+1}\mathbf{\Sigma}_\star^{1/2} - \mathbf{\Sigma}_\star\|_2 \\
&\le \frac{\|\mathbf{R}_{k+1}(\mathbf{Q}_k^{-\top} - \mathbf{Q}_{k+1}^{-\top})\mathbf{\Sigma}_\star^{1/2}\|_2}{1 - \|(\mathbf{R}_{k+1}\mathbf{Q}_{k+1}^{-\top} - \mathbf{R}_\star)\mathbf{\Sigma}_\star^{-1/2}\|_2} \\
&\le \frac{\|(\mathbf{R}_{k+1}\mathbf{Q}_k^{-\top} - \mathbf{R}_\star)\mathbf{\Sigma}_\star^{1/2}\|_2 + \|(\mathbf{R}_{k+1}\mathbf{Q}_{k+1}^{-\top} - \mathbf{R}_\star)\mathbf{\Sigma}_\star^{1/2}\|_2}{1 - \|(\mathbf{R}_{k+1}\mathbf{Q}_{k+1}^{-\top} - \mathbf{R}_\star)\mathbf{\Sigma}_\star^{-1/2}\|_2} \\
&\le \frac{2\varepsilon_0\tau^{k+1}}{1 - \varepsilon_0\tau^{k+1}}\sigma_r(\mathbf{X}_\star) \\
&\le \frac{2\varepsilon_0}{1 - \varepsilon_0}\sigma_r(\mathbf{X}_\star),
\end{aligned}$$

provided $\tau = 1 - 0.6\eta < 1$. Similarly, one can see

$$\|\mathbf{\Sigma}_\star^{1/2}\mathbf{Q}_k^\top(\mathbf{Q}_{k+1} - \mathbf{Q}_k)^{-\top}\mathbf{\Sigma}_\star^{1/2}\|_2 \le \frac{2\varepsilon_0}{1 - \varepsilon_0}\sigma_r(\mathbf{X}_\star).$$

This finishes the proof. $\qquad\square$

Notice that Lemma 9 will be only be used in the proof of Lemma 12. In the meantime, the assumption of Lemma 9 is verified in (16) (see the proof of Lemma 11).

**Lemma 10.** *If*

$$\text{dist}(\mathbf{L}_k, \mathbf{R}_k; \mathbf{L}_\star, \mathbf{R}_\star) \le \varepsilon_0\tau^k\sigma_r(\mathbf{X}_\star),$$
$$\|\mathbf{\Delta}_L\mathbf{\Sigma}_\star^{1/2}\|_{2,\infty} \vee \|\mathbf{\Delta}_R\mathbf{\Sigma}_\star^{1/2}\|_{2,\infty} \le \sqrt{\frac{\mu r}{n}}\tau^k\sigma_r(\mathbf{X}_\star),$$

*then*

$$\|\mathbf{X}_\star - \mathbf{X}_k\|_\infty \le 3\frac{\mu r}{n}\tau^k\sigma_r(\mathbf{X}_\star).$$

*Proof.* Firstly, by Assumption 1 and the assumption of this lemma, we have

$$\begin{aligned}
\|\mathbf{R}_\natural\mathbf{\Sigma}_\star^{-1/2}\|_{2,\infty} &\le \|\mathbf{\Delta}_R\mathbf{\Sigma}_\star^{1/2}\|_{2,\infty}\|\mathbf{\Sigma}_\star^{-1}\|_2 + \|\mathbf{L}_\star\mathbf{\Sigma}_\star^{-1/2}\|_{2,\infty} \\
&\le (\tau^k + 1)\sqrt{\frac{\mu r}{n}} \quad \le 2\sqrt{\frac{\mu r}{n}},
\end{aligned}$$

given $\tau = 1 - 0.6\eta < 1$. Moreover, one can see

$$\begin{aligned}
\|\mathbf{X}_\star - \mathbf{X}_k\|_\infty = \|\mathbf{\Delta}_L\mathbf{R}_\natural^\top + \mathbf{L}_\star\mathbf{\Delta}_R^\top\|_\infty &\le \|\mathbf{\Delta}_L\mathbf{R}_\natural^\top\|_\infty + \|\mathbf{L}_\star\mathbf{\Delta}_R^\top\|_\infty \\
&\le \|\mathbf{\Delta}_L\mathbf{\Sigma}_\star^{1/2}\|_{2,\infty}\|\mathbf{R}_\natural\mathbf{\Sigma}_\star^{-1/2}\|_{2,\infty} + \|\mathbf{L}_\star\mathbf{\Sigma}_\star^{-1/2}\|_{2,\infty}\|\mathbf{\Delta}_R\mathbf{\Sigma}_\star^{1/2}\|_{2,\infty} \\
&\le \left(2\sqrt{\frac{\mu r}{n}} + \sqrt{\frac{\mu r}{n}}\right)\sqrt{\frac{\mu r}{n}}\tau^k\sigma_r(\mathbf{X}_\star) \\
&= 3\frac{\mu r}{n}\tau^k\sigma_r(\mathbf{X}_\star).
\end{aligned}$$

This finishes the proof. $\qquad\square$

### A.3 Proof of local linear convergence

We will show the local convergence of the proposed algorithm by first proving the claims stand at $(k+1)$-th iteration if they stand at $k$-th iteration.

**Lemma 11.** *If*

$$\text{dist}(\boldsymbol{L}_k, \boldsymbol{R}_k; \boldsymbol{L}_\star, \boldsymbol{R}_\star) \leq \varepsilon_0 \tau^k \sigma_r(\boldsymbol{X}_\star),$$

$$\|\boldsymbol{\Delta}_L \boldsymbol{\Sigma}_\star^{1/2}\|_{2,\infty} \vee \|\boldsymbol{\Delta}_R \boldsymbol{\Sigma}_\star^{1/2}\|_{2,\infty} \leq \sqrt{\frac{\mu r}{n}} \tau^k \sigma_r(\boldsymbol{X}_\star),$$

*then*

$$\text{dist}(\boldsymbol{L}_{k+1}, \boldsymbol{R}_{k+1}; \boldsymbol{L}_\star, \boldsymbol{R}_\star) \leq \varepsilon_0 \tau^{k+1} \sigma_r(\boldsymbol{X}_\star).$$

*Proof.* Since $\boldsymbol{Q}_{k+1}$ is the optimal alignment matrix between $(\boldsymbol{L}_{k+1}, \boldsymbol{R}_{k+1})$ and $(\boldsymbol{L}_\star, \boldsymbol{R}_\star)$, so

$$\text{dist}^2(\boldsymbol{L}_{k+1}, \boldsymbol{R}_{k+1}; \boldsymbol{L}_\star, \boldsymbol{R}_\star) := \|(\boldsymbol{L}_{k+1}\boldsymbol{Q}_{k+1} - \boldsymbol{L}_\star)\boldsymbol{\Sigma}_\star^{1/2}\|_F^2 + \|(\boldsymbol{R}_{k+1}\boldsymbol{Q}_{k+1}^{-\top} - \boldsymbol{R}_\star)\boldsymbol{\Sigma}_\star^{1/2}\|_F^2$$

$$\leq \|(\boldsymbol{L}_{k+1}\boldsymbol{Q}_k - \boldsymbol{L}_\star)\boldsymbol{\Sigma}_\star^{1/2}\|_F^2 + \|(\boldsymbol{R}_{k+1}\boldsymbol{Q}_k^{-\top} - \boldsymbol{R}_\star)\boldsymbol{\Sigma}_\star^{1/2}\|_F^2$$

We will focus on bounding the first term in this proof, and the second term can be bounded similarly. Note that $\boldsymbol{L}_\natural \boldsymbol{R}_\natural^\top - \boldsymbol{X}_\star = \boldsymbol{\Delta}_L \boldsymbol{R}_\natural^\top + \boldsymbol{L}_\star \boldsymbol{\Delta}_R^\top$. We have

$$\boldsymbol{L}_{k+1}\boldsymbol{Q}_k - \boldsymbol{L}_\star = \boldsymbol{L}_\natural - \eta(\boldsymbol{L}_\natural \boldsymbol{R}_\natural^\top - \boldsymbol{X}_\star + \boldsymbol{S}_{k+1} - \boldsymbol{S}_\star)\boldsymbol{R}_\natural(\boldsymbol{R}_\natural^\top \boldsymbol{R}_\natural)^{-1} - \boldsymbol{L}_\star$$

$$= \boldsymbol{\Delta}_L - \eta(\boldsymbol{L}_\natural \boldsymbol{R}_\natural^\top - \boldsymbol{X}_\star)\boldsymbol{R}_\natural(\boldsymbol{R}_\natural^\top \boldsymbol{R}_\natural)^{-1} - \eta\boldsymbol{\Delta}_S \boldsymbol{R}_\natural(\boldsymbol{R}_\natural^\top \boldsymbol{R}_\natural)^{-1}$$

$$= (1-\eta)\boldsymbol{\Delta}_L - \eta\boldsymbol{L}_\star \boldsymbol{\Delta}_R^\top \boldsymbol{R}_\natural(\boldsymbol{R}_\natural^\top \boldsymbol{R}_\natural)^{-1} - \eta\boldsymbol{\Delta}_S \boldsymbol{R}_\natural(\boldsymbol{R}_\natural^\top \boldsymbol{R}_\natural)^{-1}. \qquad (15)$$

Thus,

$$\|(\boldsymbol{L}_{k+1}\boldsymbol{Q}_k - \boldsymbol{L}_\star)\boldsymbol{\Sigma}_\star^{1/2}\|_F^2$$
$$= \|(1-\eta)\boldsymbol{\Delta}_L \boldsymbol{\Sigma}_\star^{1/2} - \eta\boldsymbol{L}_\star \boldsymbol{\Delta}_R^\top \boldsymbol{R}_\natural(\boldsymbol{R}_\natural^\top \boldsymbol{R}_\natural)^{-1}\boldsymbol{\Sigma}_\star^{1/2}\|_F^2 - 2\eta(1-\eta)\text{tr}\left(\boldsymbol{\Delta}_S \boldsymbol{R}_\natural(\boldsymbol{R}_\natural^\top \boldsymbol{R}_\natural)^{-1}\boldsymbol{\Sigma}_\star \boldsymbol{\Delta}_L^\top\right)$$
$$+ 2\eta^2 \text{tr}\left(\boldsymbol{\Delta}_S \boldsymbol{R}_\natural(\boldsymbol{R}_\natural^\top \boldsymbol{R}_\natural)^{-1}\boldsymbol{\Sigma}_\star(\boldsymbol{R}_\natural^\top \boldsymbol{R}_\natural)^{-1}\boldsymbol{R}_\natural^\top \boldsymbol{\Delta}_R \boldsymbol{L}_\star^\top\right) + \eta^2\|\boldsymbol{\Delta}_S \boldsymbol{R}_\natural(\boldsymbol{R}_\natural^\top \boldsymbol{R}_\natural)^{-1}\boldsymbol{\Sigma}_\star^{1/2}\|_F^2$$
$$:= \mathfrak{R}_1 - \mathfrak{R}_2 + \mathfrak{R}_3 + \mathfrak{R}_4$$

**Bound of $\mathfrak{R}_1$.** The component $\mathfrak{R}_1$ here is identical to $\mathfrak{R}_1$ in [12, Section D.1.1], and the bound of this term was shown therein. We will clear this bound further by applying Lemma 7, that is,

$$\mathfrak{R}_1 \leq \left((1-\eta)^2 + \frac{2\varepsilon_0}{1-\varepsilon_0}\eta(1-\eta)\right)\|\boldsymbol{\Delta}_L \boldsymbol{\Sigma}_\star^{1/2}\|_F^2 + \frac{2\varepsilon_0 + \varepsilon_0^2}{(1-\varepsilon_0)^2}\eta^2\|\boldsymbol{\Delta}_R \boldsymbol{\Sigma}_\star^{1/2}\|_F^2$$

$$\leq (1-\eta)^2\|\boldsymbol{\Delta}_L \boldsymbol{\Sigma}_\star^{1/2}\|_F^2 + \left((1-\eta)\frac{2\varepsilon_0^3}{1-\varepsilon_0} + \eta\frac{2\varepsilon_0^3 + \varepsilon_0^4}{(1-\varepsilon_0)^2}\right)\eta\tau^{2k}\sigma_r^2(\boldsymbol{X}_\star).$$

**Bound of $\mathfrak{R}_2$.** Lemma 5 implies $\boldsymbol{\Delta}_S = \boldsymbol{S}_{k+1} - \boldsymbol{S}_\star$ is an $\alpha$-sparse matrix. Thus, by Lemmata 6, 7, 8, 5, and 10, we have

$$|\text{tr}\left(\boldsymbol{\Delta}_S \boldsymbol{R}_\natural(\boldsymbol{R}_\natural^\top \boldsymbol{R}_\natural)^{-1}\boldsymbol{\Sigma}_\star \boldsymbol{\Delta}_L^\top\right)| \leq \|\boldsymbol{\Delta}_S\|_2\|\boldsymbol{R}_\natural(\boldsymbol{R}_\natural^\top \boldsymbol{R}_\natural)^{-1}\boldsymbol{\Sigma}_\star \boldsymbol{\Delta}_L^\top\|_*$$

$$\leq \alpha n\sqrt{r}\|\boldsymbol{\Delta}_S\|_\infty\|\boldsymbol{R}_\natural(\boldsymbol{R}_\natural^\top \boldsymbol{R}_\natural)^{-1}\boldsymbol{\Sigma}_\star \boldsymbol{\Delta}_L^\top\|_F$$

$$\leq 2\alpha n\sqrt{r}\|\boldsymbol{X}_k - \boldsymbol{X}_\star\|_\infty\|\boldsymbol{R}_\natural(\boldsymbol{R}_\natural^\top \boldsymbol{R}_\natural)^{-1}\boldsymbol{\Sigma}_\star^{1/2}\|_2\|\boldsymbol{\Delta}_L \boldsymbol{\Sigma}_\star^{1/2}\|_F$$

$$\leq 6\alpha\mu r^{1.5}\tau^{2k}\frac{\varepsilon_0}{1-\varepsilon_0}\sigma_r^2(\boldsymbol{X}_\star).$$

Hence,

$$|\mathfrak{R}_2| \leq 12\eta(1-\eta)\alpha\mu r^{1.5}\tau^{2k}\frac{\varepsilon_0}{1-\varepsilon_0}\sigma_r^2(\boldsymbol{X}_\star).$$

**Bound of $\mathfrak{R}_3$.** Similar to $\mathfrak{R}_2$, we have

$$|\operatorname{tr}(\boldsymbol{\Delta}_S \boldsymbol{R}_\natural (\boldsymbol{R}_\natural^\top \boldsymbol{R}_\natural)^{-1}\boldsymbol{\Sigma}_\star(\boldsymbol{R}_\natural^\top \boldsymbol{R}_\natural)^{-1}\boldsymbol{R}_\natural^\top \boldsymbol{\Delta}_R \boldsymbol{L}_\star^\top)|$$

$$\leq \|\boldsymbol{\Delta}_S\|_2\|\boldsymbol{R}_\natural(\boldsymbol{R}_\natural^\top \boldsymbol{R}_\natural)^{-1}\boldsymbol{\Sigma}_\star(\boldsymbol{R}_\natural^\top \boldsymbol{R}_\natural)^{-1}\boldsymbol{R}_\natural^\top \boldsymbol{\Delta}_R \boldsymbol{L}_\star^\top\|_*$$

$$\leq \alpha n\sqrt{r}\|\boldsymbol{\Delta}_S\|_\infty\|\boldsymbol{R}_\natural(\boldsymbol{R}_\natural^\top \boldsymbol{R}_\natural)^{-1}\boldsymbol{\Sigma}_\star(\boldsymbol{R}_\natural^\top \boldsymbol{R}_\natural)^{-1}\boldsymbol{R}_\natural^\top \boldsymbol{\Delta}_R \boldsymbol{L}_\star^\top\|_{\mathrm{F}}$$

$$\leq \alpha n\sqrt{r}\|\boldsymbol{\Delta}_S\|_\infty\|\boldsymbol{R}_\natural(\boldsymbol{R}_\natural^\top \boldsymbol{R}_\natural)^{-1}\boldsymbol{\Sigma}_\star^{1/2}\|_2^2\|\boldsymbol{\Delta}_R \boldsymbol{L}_\star^\top\|_{\mathrm{F}}$$

$$\leq 2\alpha n\sqrt{r}\|\boldsymbol{X}_k - \boldsymbol{X}_\star\|_\infty\|\boldsymbol{R}_\natural(\boldsymbol{R}_\natural^\top \boldsymbol{R}_\natural)^{-1}\boldsymbol{\Sigma}_\star^{1/2}\|_2^2\|\boldsymbol{\Delta}_R \boldsymbol{\Sigma}_\star^{1/2}\|_{\mathrm{F}}\|\boldsymbol{U}_\star\|_2$$

$$\leq 6\alpha\mu r^{1.5}\tau^{2k}\frac{\varepsilon_0}{(1-\varepsilon_0)^2}\sigma_r^2(\boldsymbol{X}_\star).$$

Hence,

$$|\mathfrak{R}_3| \leq 12\eta^2\alpha\mu r^{1.5}\tau^{2k}\frac{\varepsilon_0}{(1-\varepsilon_0)^2}\sigma_r^2(\boldsymbol{X}_\star).$$

**Bound of $\mathfrak{R}_4$.**

$$\|\boldsymbol{\Delta}_S \boldsymbol{R}_\natural(\boldsymbol{R}_\natural^\top \boldsymbol{R}_\natural)^{-1}\boldsymbol{\Sigma}_\star^{1/2}\|_{\mathrm{F}}^2 \leq r\|\boldsymbol{\Delta}_S \boldsymbol{R}_\natural(\boldsymbol{R}_\natural^\top \boldsymbol{R}_\natural)^{-1}\boldsymbol{\Sigma}_\star^{1/2}\|_2^2$$

$$\leq r\|\boldsymbol{\Delta}_S\|_2^2\|\boldsymbol{R}_\natural(\boldsymbol{R}_\natural^\top \boldsymbol{R}_\natural)^{-1}\boldsymbol{\Sigma}_\star^{1/2}\|_2^2$$

$$\leq 4\alpha^2 n^2 r\|\boldsymbol{X}_k - \boldsymbol{X}_\star\|_\infty^2\|\boldsymbol{R}_\natural(\boldsymbol{R}_\natural^\top \boldsymbol{R}_\natural)^{-1}\boldsymbol{\Sigma}_\star^{1/2}\|_2^2$$

$$\leq 36\alpha^2\mu^2 r^3\tau^{2k}\frac{1}{(1-\varepsilon_0)^2}\sigma_r^2(\boldsymbol{X}_\star).$$

Hence,

$$\mathfrak{R}_4 \leq 36\eta^2\alpha^2\mu^2 r^3\tau^{2k}\frac{1}{(1-\varepsilon_0)^2}\sigma_r^2(\boldsymbol{X}_\star).$$

Combine all the bounds together, we have

$$\|(\boldsymbol{L}_{k+1}\boldsymbol{Q}_k - \boldsymbol{L}_\star)\boldsymbol{\Sigma}_\star^{1/2}\|_{\mathrm{F}}^2$$

$$\leq (1-\eta)^2\|\boldsymbol{\Delta}_L\boldsymbol{\Sigma}_\star^{1/2}\|_{\mathrm{F}}^2 + \left((1-\eta)\frac{2\varepsilon_0^3}{1-\varepsilon_0} + \eta\frac{2\varepsilon_0^3+\varepsilon_0^4}{(1-\varepsilon_0)^2}\right)\eta\tau^{2k}\sigma_r^2(\boldsymbol{X}_\star)$$

$$+ 12\eta(1-\eta)\alpha\mu r^{1.5}\tau^{2k}\frac{\varepsilon_0}{1-\varepsilon_0}\sigma_r^2(\boldsymbol{X}_\star)$$

$$+ 12\eta^2\alpha\mu r^{1.5}\tau^{2k}\frac{\varepsilon_0}{(1-\varepsilon_0)^2}\sigma_r^2(\boldsymbol{X}_\star)$$

$$+ 36\alpha^2\mu^2 r^3\tau^{2k}\frac{1}{(1-\varepsilon_0)^2}\sigma_r^2(\boldsymbol{X}_\star),$$

and a similar bound can be computed for $\|(\boldsymbol{R}_{k+1}\boldsymbol{Q}_k^{-\top} - \boldsymbol{R}_\star)\boldsymbol{\Sigma}_\star^{1/2}\|_{\mathrm{F}}^2$. Add together, we have

$$\operatorname{dist}^2(\boldsymbol{L}_{k+1}, \boldsymbol{R}_{k+1}; \boldsymbol{L}_\star, \boldsymbol{R}_\star)$$

$$\leq \|(\boldsymbol{L}_{k+1}\boldsymbol{Q}_k - \boldsymbol{L}_\star)\boldsymbol{\Sigma}_\star^{1/2}\|_{\mathrm{F}}^2 + \|(\boldsymbol{R}_{k+1}\boldsymbol{Q}_k^{-\top} - \boldsymbol{R}_\star)\boldsymbol{\Sigma}_\star^{1/2}\|_{\mathrm{F}}^2$$

$$\leq (1-\eta)^2\left(\|\boldsymbol{\Delta}_L\boldsymbol{\Sigma}_\star^{1/2}\|_{\mathrm{F}}^2 + \|\boldsymbol{\Delta}_R\boldsymbol{\Sigma}_\star^{1/2}\|_{\mathrm{F}}^2\right) + 2\left((1-\eta)\frac{2\varepsilon_0^3}{1-\varepsilon_0} + \eta\frac{2\varepsilon_0^3+\varepsilon_0^4}{(1-\varepsilon_0)^2}\right)\eta\tau^{2k}\sigma_r^2(\boldsymbol{X}_\star)$$

$$+ 24\eta(1-\eta)\alpha\mu r^{1.5}\tau^{2k}\frac{\varepsilon_0}{1-\varepsilon_0}\sigma_r^2(\boldsymbol{X}_\star)$$

$$+ 24\eta^2\alpha\mu r^{1.5}\tau^{2k}\frac{\varepsilon_0}{(1-\varepsilon_0)^2}\sigma_r^2(\boldsymbol{X}_\star)$$

$$+ 72\alpha^2\mu^2 r^3\tau^{2k}\frac{1}{(1-\varepsilon_0)^2}\sigma_r^2(\boldsymbol{X}_\star)$$

$$\leq \left((1-\eta)^2 + 2\left((1-\eta)\frac{2\varepsilon_0}{1-\varepsilon_0} + \eta\frac{2\varepsilon_0+\varepsilon_0^2}{(1-\varepsilon_0)^2}\right)\eta + 24\eta(1-\eta)\alpha\mu r^{1.5}\frac{1}{\varepsilon_0(1-\varepsilon_0)}\right.$$

$$+ 24\eta^2\alpha\mu r^{1.5}\frac{1}{\varepsilon_0(1-\varepsilon_0)^2} + 72\alpha^2\mu^2 r^3\frac{1}{\varepsilon_0^2(1-\varepsilon_0)^2}\Bigg)\varepsilon_0^2\tau^{2k}\sigma_r^2(\boldsymbol{X}_\star)$$

$$\leq (1-0.6\eta)^2\varepsilon_0^2\tau^{2k}\sigma_r^2(\boldsymbol{X}_\star), \tag{16}$$

where we use the fact $\|\boldsymbol{\Delta}_L\boldsymbol{\Sigma}_\star^{1/2}\|_F^2 + \|\boldsymbol{\Delta}_R\boldsymbol{\Sigma}_\star^{1/2}\|_F^2 =: \mathrm{dist}^2(\boldsymbol{L}_k, \boldsymbol{R}_k; \boldsymbol{L}_\star, \boldsymbol{R}_\star) \leq \varepsilon_0^2\tau^{2k}\sigma_r^2(\boldsymbol{X}_\star)$ in the second step, and the last step use $\varepsilon_0 = 0.02$, $\alpha \leq \frac{1}{10^4\mu r^{1.5}}$, and $\frac{1}{4} \leq \eta \leq \frac{8}{9}$. The proof is finished by substituting $\tau = 1 - 0.6\eta$.

$\square$

**Lemma 12.** *If*

$$\mathrm{dist}(\boldsymbol{L}_k, \boldsymbol{R}_k; \boldsymbol{L}_\star, \boldsymbol{R}_\star) \leq \varepsilon_0\tau^k\sigma_r(\boldsymbol{X}_\star),$$

$$\|\boldsymbol{\Delta}_L\boldsymbol{\Sigma}_\star^{1/2}\|_{2,\infty} \vee \|\boldsymbol{\Delta}_R\boldsymbol{\Sigma}_\star^{1/2}\|_{2,\infty} \leq \sqrt{\frac{\mu r}{n}}\tau^k\sigma_r(\boldsymbol{X}_\star),$$

*then*

$$\|(\boldsymbol{L}_{k+1}\boldsymbol{Q}_{k+1} - \boldsymbol{L}_\star)\boldsymbol{\Sigma}_\star^{1/2}\|_{2,\infty} \vee \|(\boldsymbol{R}_{k+1}\boldsymbol{Q}_{k+1}^{-\top} - \boldsymbol{R}_\star)\boldsymbol{\Sigma}_\star^{1/2}\|_{2,\infty} \leq \sqrt{\frac{\mu r}{n}}\tau^{k+1}\sigma_r(\boldsymbol{X}_\star).$$

*Proof.* Using (15) again, we have

$$\|(\boldsymbol{L}_{k+1}\boldsymbol{Q}_k - \boldsymbol{L}_\star)\boldsymbol{\Sigma}_\star^{1/2}\|_{2,\infty}$$

$$\leq (1-\eta)\|\boldsymbol{\Delta}_L\boldsymbol{\Sigma}_\star^{1/2}\|_{2,\infty} + \eta\|\boldsymbol{L}_\star\boldsymbol{\Delta}_R^\top\boldsymbol{R}_\natural(\boldsymbol{R}_\natural^\top\boldsymbol{R}_\natural)^{-1}\boldsymbol{\Sigma}_\star^{1/2}\|_{2,\infty} + \eta\|\boldsymbol{\Delta}_S\boldsymbol{R}_\natural(\boldsymbol{R}_\natural^\top\boldsymbol{R}_\natural)^{-1}\boldsymbol{\Sigma}_\star^{1/2}\|_{2,\infty}$$

$$:= \mathfrak{T}_1 + \mathfrak{T}_2 + \mathfrak{T}_3.$$

**Bound of $\mathfrak{T}_1$.** $\mathfrak{T}_1 \leq (1-\eta)\sqrt{\frac{\mu r}{n}}\tau^k\sigma_r(\boldsymbol{X}_\star)$ is directly followed by the assumption of this lemma.

**Bound of $\mathfrak{T}_2$.** Assumption 1 implies $\|\boldsymbol{L}_\star\boldsymbol{\Sigma}_\star^{-1/2}\|_{2,\infty} \leq \sqrt{\frac{\mu r}{n}}$, Lemma 7 implies $\|\boldsymbol{\Delta}_R\boldsymbol{\Sigma}_\star^{1/2}\|_2 \leq \tau^k\varepsilon_0$, and Lemma 8 implies Together, we have

$$\mathfrak{T}_2 \leq \eta\|\boldsymbol{L}_\star\boldsymbol{\Sigma}_\star^{-1/2}\|_{2,\infty}\|\boldsymbol{\Delta}_R\boldsymbol{\Sigma}_\star^{1/2}\|_2\|\boldsymbol{R}_\natural(\boldsymbol{R}_\natural^\top\boldsymbol{R}_\natural)^{-1}\boldsymbol{\Sigma}_\star^{1/2}\|_2$$

$$\leq \eta\frac{\varepsilon_0}{1-\varepsilon_0}\sqrt{\frac{\mu r}{n}}\tau^k\sigma_r(\boldsymbol{X}_\star).$$

**Bound of $\mathfrak{T}_3$.** By Lemma 5, $\mathrm{supp}(\boldsymbol{\Delta}_S) \subseteq \mathrm{supp}(\boldsymbol{S}_\star)$, which implies that $\boldsymbol{\Delta}_S$ is an $\alpha$-sparse matrix. Thus, by Lemma 6, we get

$$\mathfrak{T}_3 \leq \eta\|\boldsymbol{\Delta}_S\|_{2,\infty}\|\boldsymbol{R}_\natural(\boldsymbol{R}_\natural^\top\boldsymbol{R}_\natural)^{-1}\boldsymbol{\Sigma}_\star^{1/2}\|_2$$

$$\leq \eta\frac{\sqrt{\alpha n}}{1-\varepsilon_0}\|\boldsymbol{\Delta}_S\|_\infty$$

$$\leq 2\eta\frac{\sqrt{\alpha n}}{1-\varepsilon_0}\|\boldsymbol{X}_\star - \boldsymbol{X}_k\|_\infty$$

$$\leq 6\eta\frac{\sqrt{\alpha\mu r}}{1-\varepsilon_0}\sqrt{\frac{\mu r}{n}}\tau^k\sigma_r(\boldsymbol{X}_\star).$$

where the last two steps use Lemmata 5 and 10 respectively. Put together, we obtain

$$\|(\boldsymbol{L}_{k+1}\boldsymbol{Q}_k - \boldsymbol{L}_\star)\boldsymbol{\Sigma}_\star^{1/2}\|_{2,\infty} \leq \mathfrak{T}_1 + \mathfrak{T}_2 + \mathfrak{T}_3$$

$$\leq \left(1 - \eta + \eta\frac{\varepsilon_0}{1-\varepsilon_0} + 6\eta\frac{\sqrt{\alpha\mu r}}{1-\varepsilon_0}\right)\sqrt{\frac{\mu r}{n}}\tau^k\sigma_r(\boldsymbol{X}_\star)$$

$$\leq \left(1 - \eta\left(1 - \frac{\varepsilon_0}{1-\varepsilon_0} - 6\frac{\sqrt{\alpha\mu r}}{1-\varepsilon_0}\right)\right)\sqrt{\frac{\mu r}{n}}\tau^k\sigma_r(\boldsymbol{X}_\star). \tag{17}$$

In addition, we also have

$$\|(\boldsymbol{L}_{k+1}\boldsymbol{Q}_k - \boldsymbol{L}_\star)\boldsymbol{\Sigma}_\star^{-1/2}\|_{2,\infty} \leq \left(1 - \eta\left(1 - \frac{\varepsilon_0}{1-\varepsilon_0} - 6\frac{\sqrt{\alpha\mu r}}{1-\varepsilon_0}\right)\right)\sqrt{\frac{\mu r}{n}}\tau^k. \tag{18}$$

**Bound with $Q_{k+1}$.** Note that $Q$'s are the best align matrices under Frobenius norm but this is not necessary true under $\ell_{2,\infty}$ norm. So we must show the bound of $\|(L_{k+1}Q_{k+1} - L_\star)\Sigma_\star^{1/2}\|_{2,\infty}$ directly. Note that $Q_{k+1}$ does exist, according to Lemmata 11 and 2. Applying (17), (18) and Lemma 9, we have

$$\|(L_{k+1}Q_{k+1} - L_\star)\Sigma_\star^{1/2}\|_{2,\infty}$$
$$\leq \|(L_{k+1}Q_k - L_\star)\Sigma_\star^{1/2}\|_{2,\infty} + \|L_{k+1}(Q_{k+1} - Q_k)\Sigma_\star^{1/2}\|_{2,\infty}$$
$$= \|(L_{k+1}Q_k - L_\star)\Sigma_\star^{1/2}\|_{2,\infty} + \|L_{k+1}Q_k\Sigma_\star^{-1/2}\Sigma_\star^{1/2}Q_k^{-1}(Q_{k+1} - Q_k)\Sigma_\star^{1/2}\|_{2,\infty}$$
$$\leq \|(L_{k+1}Q_k - L_\star)\Sigma_\star^{1/2}\|_{2,\infty} + \|L_{k+1}Q_k\Sigma_\star^{-1/2}\|_{2,\infty}\|\Sigma_\star^{1/2}Q_k^{-1}(Q_{k+1} - Q_k)\Sigma_\star^{1/2}\|_2$$
$$\leq \|(L_{k+1}Q_k - L_\star)\Sigma_\star^{1/2}\|_{2,\infty}$$
$$\quad + \left(\|(L_{k+1}Q_k - L_\star)\Sigma_\star^{-1/2}\|_{2,\infty} + \|L_\star\Sigma_\star^{-1/2}\|_{2,\infty}\right)\|\Sigma_\star^{1/2}Q_k^{-1}(Q_{k+1} - Q_k)\Sigma_\star^{1/2}\|_2$$
$$\leq \left(1 - \eta\left(1 - \frac{\varepsilon_0}{1 - \varepsilon_0} - 6\frac{\sqrt{\alpha\mu r}}{1 - \varepsilon_0}\right) + \frac{2\varepsilon_0}{1 - \varepsilon_0}\left(2 - \eta\left(1 - \frac{\varepsilon_0}{1 - \varepsilon_0} - 6\frac{\sqrt{\alpha\mu r}}{1 - \varepsilon_0}\right)\right)\right)$$
$$\sqrt{\frac{\mu r}{n}}\tau^k\sigma_r(X_\star)$$
$$\leq (1 - 0.6\eta)\sqrt{\frac{\mu r}{n}}\tau^k\sigma_r(X_\star),$$

where the last step use $\varepsilon_0 = 0.02$, $\alpha \leq \frac{1}{10^4\mu r^{1.5}}$, and $\frac{1}{4} \leq \eta \leq \frac{8}{9}$. Similar result can be computed for $\|(R_{k+1}Q_{k+1}^{-\top} - R_\star)\Sigma_\star^{1/2}\|_{2,\infty}$. The proof is finished by substituting $\tau = 1 - 0.6\eta$. $\qquad\square$

Now we have all the ingredients for proving the theorem of local linear convergence, i.e., Theorem 3.

*Proof of Theorem 3.* This proof is done by induction.

**Base case.** Since $\tau^0 = 1$, the assumed initial conditions satisfy the base case at $k = 0$.

**Induction step.** At the $k$-th iteration, we assume the conditions

$$\text{dist}(L_k, R_k; L_\star, R_\star) \leq \varepsilon_0\tau^k\sigma_r(X_\star),$$
$$\|(L_kQ_k - L_\star)\Sigma_\star^{1/2}\|_{2,\infty} \vee \|(R_kQ_k^{-\top} - R_\star)\Sigma_\star^{1/2}\|_{2,\infty} \leq \sqrt{\frac{\mu r}{n}}\tau^k\sigma_r(X_\star)$$

hold, then by Lemmata 11 and 12,

$$\text{dist}(L_{k+1}, R_{k+1}; L_\star, R_\star) \leq \varepsilon_0\tau^{k+1}\sigma_r(X_\star),$$
$$\|(L_{k+1}Q_{k+1} - L_\star)\Sigma_\star^{1/2}\|_{2,\infty} \vee \|(R_{k+1}Q_{k+1}^{-\top} - R_\star)\Sigma_\star^{1/2}\|_{2,\infty} \leq \sqrt{\frac{\mu r}{n}}\tau^{k+1}\sigma_r(X_\star)$$

also hold. This finishes the proof. $\qquad\square$

## A.4 Proof of guaranteed initialization

Now we show the outputs of the initialization step in Algorithm 1 satisfy the initial conditions required by Theorem 3.

*Proof of Theorem 4.* Firstly, by Assumption 1, we obtain

$$\|X_\star\|_\infty \leq \|U_\star\|_{2,\infty}\|\Sigma_\star\|_2\|V_\star\|_{2,\infty} \leq \frac{\mu r}{n}\sigma_1(X_\star).$$

Invoking Lemma 5 with $X_{-1} = 0$, we have

$$\|S_\star - S_0\|_\infty \leq 2\frac{\mu r}{n}\sigma_1(X_\star) \qquad \text{and} \qquad \text{supp}(S_0) \subseteq \text{supp}(S_\star), \tag{19}$$

which implies $S_\star - S_0$ is an $\alpha$-sparse matrix. Applying Lemma 6, we have
$$\|S_\star - S_0\|_2 \leq \alpha n \|S_\star - S_0\|_\infty \leq 2\alpha\mu r\sigma_1(X_\star) = 2\alpha\mu r\kappa\sigma_r(X_\star).$$

Since $X_0 = L_0 R_0^\top$ is the best rank-$r$ approximation of $Y - S_0$, so
$$\begin{aligned}
\|X_\star - X_0\|_2 &\leq \|X_\star - (Y - S_0)\|_2 + \|(Y - S_0) - X_0\|_2 \\
&\leq 2\|X_\star - (Y - S_0)\|_2 \\
&= 2\|S_\star - S_0\|_2 \\
&\leq 4\alpha\mu r\kappa\sigma_r(X_\star),
\end{aligned}$$

where the equality uses the definition $Y = X_\star + S_\star$. By [12, Lemma 11], we obtain
$$\begin{aligned}
\mathrm{dist}(L_0, R_0; L_\star, R_\star) &\leq \sqrt{\sqrt{2}+1}\|X_\star - X_0\|_F \\
&\leq \sqrt{(\sqrt{2}+1)2r}\|X_\star - X_0\|_2 \\
&\leq 10\alpha\mu r^{1.5}\kappa\sigma_r(X_\star),
\end{aligned}$$

where we use the fact that $X_\star - X_0$ has at most rank-$2r$. Given $\varepsilon_0 = 10c_0$ and $\alpha \leq \frac{c_0}{\mu r^{1.5}\kappa}$, our first claim
$$\mathrm{dist}(L_0, R_0; L_\star, R_\star) \leq 10c_0\sigma_r(X_\star) \tag{20}$$

is proved.

Let $\varepsilon_0 := 10c_0$. Now, we will prove the second claim:
$$\|\Delta_L \Sigma_\star^{1/2}\|_{2,\infty} \vee \|\Delta_R \Sigma_\star^{1/2}\|_{2,\infty} \leq \sqrt{\frac{\mu r}{n}}\sigma_r(X_\star)$$

where $\Delta_L := L_0 Q_0 - L_\star$ and $\Delta_R := R_0 Q_0^{-\top} - R_\star$. For ease of notation, we also denote $L_\natural = L_0 Q_0$, $R_\natural = R_0 Q_0^{-\top}$, and $\Delta_S = S_0 - S_\star$ in the rest of this proof.

We will work on $\|\Delta\Sigma_\star^{1/2}\|_{2,\infty}$ first, and $\|\Delta\Sigma_\star^{1/2}\|_{2,\infty}$ can be bounded similarly.

Since $U_0\Sigma_0 V_0^\top = \mathcal{D}_r(Y - S_0) = \mathcal{D}_r(X_\star - \Delta_S)$, so
$$\begin{aligned}
L_0 = U_0\Sigma_0^{1/2} &= (X_\star - \Delta_S)V_0\Sigma_0^{-1/2} \\
&= (X_\star - \Delta_S)R_0\Sigma_0^{-1} \\
&= (X_\star - \Delta_S)R_0(R_0^\top R_0)^{-1}.
\end{aligned}$$

Multiplying $Q_0\Sigma_\star^{1/2}$ on both sides, we have
$$\begin{aligned}
L_\natural\Sigma_\star^{1/2} = L_0 Q_0\Sigma_\star^{1/2} &= (X_\star - \Delta_S)R_0(R_0^\top R_0)^{-1}Q_0\Sigma_\star^{1/2} \\
&= (X_\star - \Delta_S)R_\natural(R_\natural^\top R_\natural)^{-1}\Sigma_\star^{1/2}.
\end{aligned}$$

Subtracting $X_\star R_\natural(R_\natural^\top R_\natural)^{-1}\Sigma_\star^{1/2}$ on both sides, we have
$$L_\natural\Sigma_\star^{1/2} - L_\star R_\star^\top R_\natural(R_\natural^\top R_\natural)^{-1}\Sigma_\star^{1/2} = (X_\star - \Delta_S)R_\natural(R_\natural^\top R_\natural)^{-1}\Sigma_\star^{1/2} - X_\star R_\natural(R_\natural^\top R_\natural)^{-1}\Sigma_\star^{1/2}$$
$$\Delta_L\Sigma_\star^{1/2} + L_\star\Delta_R^\top R_\natural(R_\natural^\top R_\natural)^{-1}\Sigma_\star^{1/2} = -\Delta_S R_\natural(R_\natural^\top R_\natural)^{-1}\Sigma_\star^{1/2},$$

where the left operand of last step uses the fact $L_\star\Sigma_\star^{1/2} = L_\star R_\natural^\top R_\natural(R_\natural^\top R_\natural)^{-1}\Sigma^{1/2}$. Thus,
$$\begin{aligned}
\|\Delta_L\Sigma_\star^{1/2}\|_{2,\infty} &\leq \|L_\star\Delta_R^\top R_\natural(R_\natural^\top R_\natural)^{-1}\Sigma_\star^{1/2}\|_{2,\infty} + \|\Delta_S R_\natural(R_\natural^\top R_\natural)^{-1}\Sigma_\star^{1/2}\|_{2,\infty} \\
&:= \mathfrak{J}_1 + \mathfrak{J}_2
\end{aligned}$$

**Bound of $\mathfrak{J}_1$.** By Assumption 1, we get
$$\begin{aligned}
\mathfrak{J}_1 &\leq \|L_\star\Sigma_\star^{-1/2}\|_{2,\infty}\|\Delta_R\Sigma_\star^{1/2}\|_2\|R_\natural(R_\natural^\top R_\natural)^{-1}\Sigma_\star^{1/2}\|_2 \\
&\leq \sqrt{\frac{\mu r}{n}}\frac{\varepsilon_0}{1-\varepsilon_0}\sigma_r(X_\star)
\end{aligned}$$

where Lemma 7 implies $\|\Delta_R\Sigma_\star^{1/2}\|_2 \leq \varepsilon_0\sigma_r(X_\star)$, and Lemma 8 implies $\|R_\natural(R_\natural^\top R_\natural)^{-1}\Sigma_\star^{1/2}\|_2 \leq \frac{1}{1-\varepsilon_0}$, given (20) holds.

**Bound of $\mathfrak{J}_2$.** (19) implies $\boldsymbol{\Delta}_S$ is $\alpha$-sparse. Moreover, by (20), Lemmata 6 and 8, we have

$$
\begin{aligned}
\mathfrak{J}_2 &\leq \|\boldsymbol{\Delta}_S\|_{1,\infty}\|\boldsymbol{R}_\natural \boldsymbol{\Sigma}_\star^{-1/2}\|_{2,\infty}\|\boldsymbol{\Sigma}_\star^{1/2}(\boldsymbol{R}_\natural^\top \boldsymbol{R}_\natural)^{-1}\boldsymbol{\Sigma}_\star^{1/2}\|_2 \\
&\leq \alpha n\|\boldsymbol{\Delta}_S\|_\infty\|\boldsymbol{R}_\natural \boldsymbol{\Sigma}_\star^{-1/2}\|_{2,\infty}\|\boldsymbol{R}_\natural(\boldsymbol{R}_\natural^\top \boldsymbol{R}_\natural)^{-1}\boldsymbol{\Sigma}_\star^{1/2}\|_2^2 \\
&\leq \alpha n\frac{2\mu r}{n}\sigma_1(\boldsymbol{X}_\star)\frac{1}{(1-\varepsilon_0)^2}\|\boldsymbol{R}_\natural \boldsymbol{\Sigma}_\star^{-1/2}\|_{2,\infty} \\
&\leq \frac{2\alpha\mu r\kappa}{(1-\varepsilon_0)^2}\left(\sqrt{\frac{\mu r}{n}}+\|\boldsymbol{\Delta}_R \boldsymbol{\Sigma}_\star^{-1/2}\|_{2,\infty}\right)\sigma_r(\boldsymbol{X}_\star)
\end{aligned}
$$

where the first step uses that $\|\boldsymbol{AB}\|_{2,\infty} \leq \|\boldsymbol{A}\|_{1,\infty}\|\boldsymbol{B}\|_{2,\infty}$ for any matrices. Note that $\|\boldsymbol{\Delta}_R \boldsymbol{\Sigma}_\star^{-1/2}\|_{2,\infty} \leq \frac{\|\boldsymbol{\Delta}_R \boldsymbol{\Sigma}_\star^{1/2}\|_{2,\infty}}{\sigma_r(\boldsymbol{X}_\star)}$. Hence,

$$
\|\boldsymbol{\Delta}_L \boldsymbol{\Sigma}_\star^{1/2}\|_{2,\infty} \leq \left(\frac{\varepsilon_0}{1-\varepsilon_0}+\frac{2\alpha\mu r\kappa}{(1-\varepsilon_0)^2}\right)\sqrt{\frac{\mu r}{n}}\sigma_r(\boldsymbol{X}_\star)+\frac{2\alpha\mu r\kappa}{(1-\varepsilon_0)^2}\|\boldsymbol{\Delta}_R \boldsymbol{\Sigma}_\star^{1/2}\|_{2,\infty},
$$

and similarly one can see

$$
\|\boldsymbol{\Delta}_R \boldsymbol{\Sigma}_\star^{1/2}\|_{2,\infty} \leq \left(\frac{\varepsilon_0}{1-\varepsilon_0}+\frac{2\alpha\mu r\kappa}{(1-\varepsilon_0)^2}\right)\sqrt{\frac{\mu r}{n}}\sigma_r(\boldsymbol{X}_\star)+\frac{2\alpha\mu r\kappa}{(1-\varepsilon_0)^2}\|\boldsymbol{\Delta}_L \boldsymbol{\Sigma}_\star^{1/2}\|_{2,\infty}.
$$

Therefore, substituting $\varepsilon_0 = 10c_0$ gives

$$
\begin{aligned}
&\|\boldsymbol{\Delta}_L \boldsymbol{\Sigma}_\star^{1/2}\|_{2,\infty} \vee \|\boldsymbol{\Delta}_R \boldsymbol{\Sigma}_\star^{1/2}\|_{2,\infty} \\
&\leq \frac{(1-\varepsilon_0)^2}{(1-\varepsilon_0)^2-2\alpha\mu r\kappa}\left(\frac{\varepsilon_0}{1-\varepsilon_0}+\frac{2\alpha\mu r\kappa}{(1-\varepsilon_0)^2}\right)\sqrt{\frac{\mu r}{n}}\sigma_r(\boldsymbol{X}_\star) \\
&\leq \frac{(1-10c_0)^2}{(1-10c_0)^2-2c_0}\left(\frac{10c_0}{1-10c_0}+\frac{2c_0}{(1-10c_0)^2}\right)\sqrt{\frac{\mu r}{n}}\sigma_r(\boldsymbol{X}_\star) \\
&\leq \sqrt{\frac{\mu r}{n}}\sigma_r(\boldsymbol{X}_\star),
\end{aligned}
$$

as long as $c_0 \leq \frac{1}{35}$. This finishes the proof. $\qquad\square$

# B  Complexity of LRPCA

We provide the breakdown of LRPCA's computational complexity:

1. Compute $\boldsymbol{L}_k \boldsymbol{R}_k^\top$: $n$-by-$r$ matrix times $r$-by-$n$ matrix—$n^2 r$ flops.
2. Compute $\boldsymbol{Y} - \boldsymbol{L}_k \boldsymbol{R}_k^\top$: $n$-by-$n$ matrix minus $n$-by-$n$ matrix—$n^2$ flops.
3. Soft-thresholding on $\boldsymbol{Y} - \boldsymbol{L}_k \boldsymbol{R}_k^\top$: one pass on $n$-by-$n$ matrix—$n^2$ flops.
4. Compute $\boldsymbol{L}_k \boldsymbol{R}_k^\top + \boldsymbol{S}_{k+1} - \boldsymbol{Y} = \boldsymbol{S}_{k+1} - (\boldsymbol{Y} - \boldsymbol{L}_k \boldsymbol{R}_k^\top)$: $n$-by-$n$ matrix minus $n$-by-$n$ matrix—$n^2$ flops.
5. Compute $\boldsymbol{R}_k^\top \boldsymbol{R}_k$: $r$-by-$n$ matrix times $n$-by-$r$ matrix—$nr^2$ flops.
6. Compute $(\boldsymbol{R}_k^\top \boldsymbol{R}_k)^{-1}$: invert a $r$-by-$r$ matrix—$\mathcal{O}(r^3)$ flops.
7. Compute $\boldsymbol{R}_k(\boldsymbol{R}_k^\top \boldsymbol{R}_k)^{-1}$: $n$-by-$r$ matrix times $r$-by-$r$ matrix—$nr^2$ flops.
8. Compute $(\boldsymbol{L}_k \boldsymbol{R}_k^\top + \boldsymbol{S}_{k+1} - \boldsymbol{Y})\cdot\boldsymbol{R}_k(\boldsymbol{R}_k^\top \boldsymbol{R}_k)^{-1}$: $n$-by-$n$ matrix times $n$-by-$r$ matrix—$n^2 r$ flops.
9. Compute $\boldsymbol{L}_{k+1} = \boldsymbol{L}_k - \zeta_{k+1}(\boldsymbol{L}_k \boldsymbol{R}_k^\top + \boldsymbol{S}_{k+1} - \boldsymbol{Y})\boldsymbol{R}_k(\boldsymbol{R}_k^\top \boldsymbol{R}_k)^{-1}$: $n$-by-$r$ matrix minus scalar times $n$-by-$r$ matrix—$2nr$ flops.
10. Repeat step 5 - 9 for computing $\boldsymbol{R}_{k+1}$—another $2nr^2 + \mathcal{O}(r^3) + n^2 r + 2nr$ flops.

In total, LRPCA costs $3n^2 r + 3n^2 + \mathcal{O}(nr^2)$ flops per iteration provided $r \ll n$. Note that we count $abc$ flops for computing an $a$-by-$b$ matrix times a $b$-by-$c$ matrix in the above complexity calculation. Some may argue that this matrix product should take $2abc$ flops. The per-iteration complexity can be rectified to $6n^2 r + 3n^2 + \mathcal{O}(nr^2)$ flops if the reader prefers the latter opinion.

## C  Additional numerical results

### C.1  Setup details

**Random instance generation.**  We follow the setup in [8, 10] to generate synthetic data. Each observation signal $\boldsymbol{Y}_\star \in \mathbb{R}^{n \times n}$ is generated by $\boldsymbol{Y}_\star = \boldsymbol{X}_\star + \boldsymbol{S}_\star$. The underlying low-rank matrix $\boldsymbol{X}_\star$ is generated with $\boldsymbol{X}_\star = \boldsymbol{L}_\star \boldsymbol{R}_\star^\top$ where $\boldsymbol{L}_\star, \boldsymbol{R}_\star \in \mathbb{R}^{n \times r}$ have elements drawn i.i.d from zero-mean Gaussian distribution with variance $1/n$. Non-zero locations of the underlying sparse matrix $\boldsymbol{S}_\star$ is uniformly and independently sampled without replacement. The magnitudes of the non-zeros of $\boldsymbol{S}_\star$ are sampled i.i.d from the uniform distribution over the interval $[-\mathbb{E}|[\boldsymbol{X}_\star]_{i,j}|, \mathbb{E}|[\boldsymbol{X}_\star]_{i,j}|]$.

**Video instances preprocessing.**  To accelerate the training process, we first change the RGB videos in the VIRAT dataset to gray videos and then downsample the videos by a fraction of $4$. All training videos are cut to sub-videos with number of frames no more than $1000$, testing videos are not cut.

**Details in training.**  In the layer-wise training phase, we adopt SGD with batch size $1$; in the parameter $(\beta, \phi)$ searching phase (i.e., RNN training), we adopt grid search with grid size $0.1$. In synthetic data experiments, the ground truth $\boldsymbol{X}_\star$ is known after each instance is generated. Thus, the training pair $(\boldsymbol{Y}, \boldsymbol{X}_\star)$ is easy to obtain. We generate a new instance in each step of SGD in the layer-wise training phase and generate $20$ instances for the grid search phase. The testing set is separately generated and consists of $50$ instances. In the video experiment, the underlying ground truth $\boldsymbol{X}_\star$ is unknown. We solve each training video with a classic RPCA algorithm [6] (without learning) to precision $10^{-5}$ and use that solution as $\boldsymbol{X}_\star$. Moreover, in synthetic data experiments, we we set $K = 10, \overline{K} = 15$; in video experiments, we set $K = 5, \overline{K} = 10$ and the underlying rank $r = 2$.

### C.2  Training time

Our training time for different matrix sizes, ranks, and outlier densities are reported in Table 4.

Table 4: Training time summary.

| Problem settings | Training time |
|---|---|
| $n = 1000, r = 5, \alpha = 0.1$ | 1208 secs |
| $n = 1000, r = 5, \alpha = 0.2$ | 1209 secs |
| $n = 1000, r = 5, \alpha = 0.3$ | 1208 secs |
| $n = 1000, r = 5, \alpha = 0.1$ | 1208 secs |
| $n = 3000, r = 5, \alpha = 0.1$ | 1615 secs |
| $n = 5000, r = 5, \alpha = 0.1$ | 2405 secs |
| $n = 1000, r = 5, \alpha = 0.1$ | 1208 secs |
| $n = 1000, r = 10, \alpha = 0.1$ | 1236 secs |
| $n = 1000, r = 15, \alpha = 0.1$ | 1249 secs |

Different from the inference time reported in the main paper, the training was done on a workstation equipped with two Nvidia RTX-3080 GPUs. Note that the training time is not proportional to the problem size due to the high concurrency of GPU computing.

### C.3  Visualizations of video background subtraction

In Figure 7, we visualize the results of LRPCA, ScaledGD and AltProj on the task of video background subtraction.

### C.4  Generalization

In this section, we study the generalization ability of our model. Specifically, we train our model on small-size and low-rank instances, and test it on instances with larger size or higher rank.

First we train a FRMNN on instances of size $1000 \times 1000$ and rank-5. This setting is denoted as the "base" setting. We only train the model once on the base setting and test it on instances with different

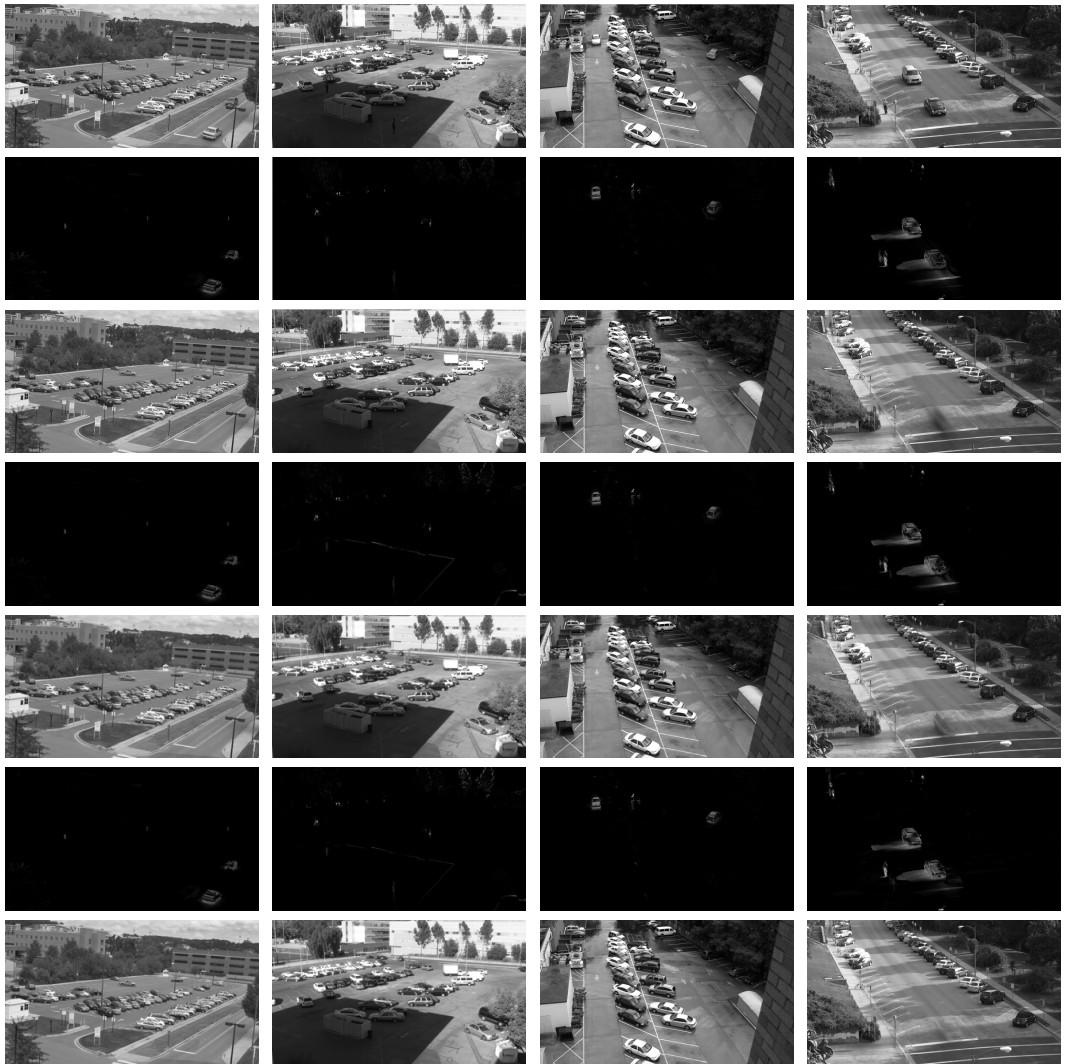

Figure 7: Video background subtraction visual results. Each column represents a selected frame from the tested videos. From left to right, they are ParkingLot1, ParkingLot2, ParkingLot3, and StreetView. The first row contains the original frames. Rows 2 and 3 are the separated foreground and background produced by LRPCA, respectively. Rows 4 and 5 are results for ScaledGD. The last two rows are results for AltProj.

settings (denoted as "target" settings). When we test a model on instances, we use the step sizes $\{\eta_k\}$ directly and scale the thresholdings $\{\zeta_k\}$ by a factor of $(n_{\text{base}}/n_{\text{target}})(r_{\text{target}}/r_{\text{base}})$ due to the $\ell_\infty$ bound estimation given in Lemma 10.

As a comparison, we also train FRMNNs individually on the target settings. The averaged iterations to achieve $10^{-4}$ on the testing set are reported in Table 5.

From Table 5, we conclude that our model has good generalization ability w.r.t. $n$ and $r$. For example, if we train and test a model both with $n = 3000, r = 5$, it takes 6 iterations; if we train a model on the base setting (i.e., $n = 1000, r = 5$) and test it with $n = 3000, r = 5$, it takes 7 iterations. Such generalization of our model works fine with slight loss of performance. That is, a model trained once on the base setting is good enough for larger size or higher rank problems from similar testing sets.

Table 5: Results for generalization test.

| Fix $r = 5$, test different $n$ | | | |
|---|---|---|---|
| Matrix size $n$ | 1000 | 3000 | 5000 |
| Iterations (Model trained on base setting) | 8 | 7 | 7 |
| Iterations (Model trained on target setting) | 8 | 6 | 5 |
| Fix $n = 1000$, test different $r$ | | | |
| Matrix rank $r$ | 5 | 10 | 15 |
| Iterations (Model trained on base setting) | 8 | 10 | 11 |
| Iterations (Model trained on target setting) | 8 | 8 | 9 |

## C.5 Analysis of trained parameters

We visualize the trained step sizes and thresholdings in Figures 8 and 9, respectively. Figure 9 demonstrates that the trained thresholdings decay in an exponential rate, which is aligned with our theoretical bound in Lemma 10. Figure 8 shows that $\eta_k$ takes larger value when $k$ is small. In other words, the algorithm goes very aggressively with large step sizes in the first several steps.

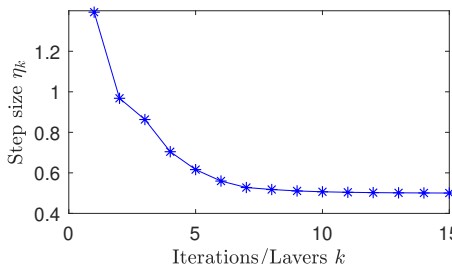

Figure 8: Trained step sizes

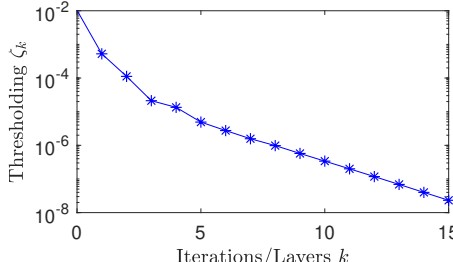

Figure 9: Trained thresholdings