# OpenReview forum: "Learned Robust PCA: A Scalable Deep Unfolding Approach for High-Dimensional Outlier Detection"
_NeurIPS.cc/2021/Conference — NeurIPS 2021 Poster_

### Official Review · Reviewer_7NwB · 2021-07-01

**Rating:** 6
**Confidence:** 4

**Summary:**

The paper propose a new technique for robust PCA that can be realised as a trainable neural network. The technique provides similar theoretical guarantees to existing methods, yet scale better to larger problems. The paper is well-written and mostly easy to follow.

**Ethical Concerns:**

I see no ethical concerns; the work is methodological/theoretical.

**Limitations And Societal Impact:**

I was satisfied with this aspect of the paper.

**Main Review:**

The paper considers robust PCA under the usual assumption of the data being an additive composition of a low-rank matrix and a sparse (outlier) matrix. This is solved with a  technique that is quite similar to ScaledGD [10], which provide a neat gradient update that is numerically well-behaved. As I see it, the main contribution of the paper is then an 'unfolding' of this method such that it can be realized as (trainable) feed-forward neural network. This allows for significant speed-ups compared to similar methods.

Positive:
-- the paper is well-written, and does a nice job of placing itself with respect to the existing literature.
-- the observation that an RNN version of the unfolded algorithm is unstable is (in many ways) unsurprising, but the straight-forward solution concatenating a feed-forward network with an RNN is neat. It's sufficiently simple that it trust it to work.
-- the theoretical guarantees are nice; it's good to see that nothing is lost when going to more scalable algorithms.
-- Theorem 1 seems quite useful as it justifies the simple soft-thresholding.

Less-positive:
-- While the paper does a nice positioning job relative to similar RPCA methods, other branches of the RPCA literature is ignored. E.g. statements such as "Nevertheless, all existing approaches call an expensive step of singular value thresholding (SVT)" (line 30-31) is only true with a highly narrow definition of RPCA. For example, methods based on Grassmann averages

  *) Scalable Robust Principal Component Analysis using Grassmann Averages
     Søren Hauberg, Aasa Feragen, Raffi Enficiaud and Michael J. Black.
     In IEEE Transactions on Pattern Analysis and Machine Intelligence (TPAMI).

  *) Intrinsic Grassmann Averages for Online Linear Robust and Nonlinear Subspace Learning
      Rudrasis Chakraborty, Liu Yang, Søren Hauberg and Baba C. Vemuri.
      In IEEE Transactions on Pattern Analysis and Machine Intelligence (TPAMI).

does not suffer from such limitations. The first of the papers also demonstrated RPCA on terabytes of data (Star Wars IV, in HD resolution), which would indicate a significantly more scalable algorithm than the one presented here. I do not think that these (and similar) existing works diminish the contribution of the present paper, but the related work should be acknowledged.

--  Related: I found it rather confusing that the paper argues that SVD is not needed by the proposed algorithm, yet, line 4 of Algorithm 1 is a call to SVD. I acknowledge that this is just *one* call to SVD where as existing methods call SVD repeatedly, but for large data sets even a single call to SVD can be computationally (and memory-wise) prohibitive.

-- As I read the paper, the proposed method is very similar to ScaledGD. It isn't entirely clear to me if there are other benefits than a faster algorithm. Computational speed is important, so I think it is fine if a speed-up is the only benefit, but I could use a more explicit discussion of this in the paper.

-- The section "More numerical experiments" could to with a rewrite. The current text states which additional experiments can be found in the supplements, but it would be better if the text also provided a high-level summary of the findings.

Other:
-- I could not find any mention of licenses in Section 4 even if this is stated in the checklist.
-- The authors did not include code, but have promised to release upon acceptance. I find this satisfactory.

**Time Spent Reviewing:**

1

---

> ### Author Response · Authors · 2021-08-10
> **Response to Reviewer 7NwB**
>
> ### Question: While the paper does a nice positioning job relative to similar RPCA methods, other branches of the RPCA literature is ignored. E.g. statements such as "Nevertheless, all existing approaches call an expensive step of singular value thresholding (SVT)" (line 30-31) is only true with a highly narrow definition of RPCA. For example, methods based on Grassmann averages [Hauberg et al. ‘16, Chakraborty et al. ‘20] do not suffer from such limitations. The first of the papers also demonstrated RPCA on terabytes of data (Star Wars IV, in HD resolution), which would indicate a significantly more scalable algorithm than the one presented here. I do not think that these (and similar) existing works diminish the contribution of the present paper, but the related work should be acknowledged.
>
> Response: Thank you for pointing this out. To clarify, the quoted statement is applied in the context of “all existing learning-based approaches for RPCA model (1)” rather than  “all RPCA approaches for any RPCA model”. This will be clarified in the revision.
>
> Thank you for the missed references. **In the camera-ready version, we are happy to add a short discussion for this line of RPCA research to the section of related work.**
>
> In particular, we will include the following sentence and make adjustments around:
>
> *When the columns of $X_\star$ are independently sampled over a zero-mean multivariate Gaussian distribution, a highly efficient approach for solving high-dimensional RPCA is via Grassmann averages [Hauberg et al. ‘14, Hauberg et al. ‘16, Chakraborty et al. ‘20].*
>
> -----------------------------------------------
>
> ### Question: I found it rather confusing that the paper argues that SVD is not needed by the proposed algorithm, yet, line 4 of Algorithm 1 is a call to SVD. I acknowledge that this is just one call to SVD where as existing methods call SVD repeatedly, but for large data sets even a single call to SVD can be computationally (and memory-wise) prohibitive.
>
> Response: Thank you for pointing this out. We use one step of **truncated SVD** in the initialization, which is much cheaper than a full SVD; that is, $O(n^2 r)$ vs. $O(n^3)$. LRPCA does require a proper initialization, but it does not have to be through SVD. For example, batch-based low rank approximation methods, such as Nystome approximation and CUR decomposition, can provide an empirically good enough initialization with high probability. While these approximations are more computational and memory efficient than truncated SVD, their stabilities lack theoretical support when outliers appear. Thus, we use the theoretically guaranteed spectral initialization, which involves one step of truncated SVD, for demonstration. We emphasize that our learning framework can apply to some other efficient initialization methods as well. **That being said, we are happy to add the above discussion as a remark in the camera-ready version.**
>
> -----------------------------------------------
>
> ### Question: As I read the paper, the proposed method is very similar to ScaledGD. It isn't entirely clear to me if there are other benefits than a faster algorithm. Computational speed is important, so I think it is fine if a speed-up is the only benefit, but I could use a more explicit discussion of this in the paper.
>
> Thank you for this valuable comment. We can list two more benefits here:
>
> 1. LRPCA’s parameters are learnable while ScaledGD’s parameters are not learnable since the sparsification operator is indifferentiable. In the other words, ScaledGD is not natively suitable for the existing learn-to-optimize framework. This benefit has been discussed in Lines 133-137.
>
> 2. LRPCA is more robust than ScaledGD when $\alpha$ is larger, i.e., more outlier tolerance. Recall that ScaledGD takes off the largest $\gamma \alpha$ fraction entries per row and per column via its sparsification operator. Let’s take the parameter $\gamma = 1.5$ for now ($\gamma$ must be strictly greater than $1$, and ScaledGD uses $\gamma=2$ for analysis). When we have only 10% outliers, the sparsification operator takes off 15% entries---not too bad.  If we have 40% outliers,  the sparsification operator takes off 60% entries---lose too many good entries. In contrast, the thresholding operator (with a properly chosen thresholding value) in LRPCA takes off no more than 40% entries, and all good entries are saved. In this sense, LRPCA preserves more good redundant information than ScaledGD, especially when more outliers appear. Hence,  it is not a surprise that LRPCA outperforms ScaledGD when $\alpha$ is larger.
>
> **We are happy to add a short discussion in the camera-ready version to stress these additional benefits.**
>
> In addition, we add the following experiment to support our claim:
>
> *We generate 10 problems for each of the outlier levels (i.e., $\alpha$) and compare the recoverability of LRPCA and ScaledGD. The test result is presented in the table below, where we can conclude that LRPCA is more robust when $\alpha$ is larger.*
>
> |$\alpha$ |	0.4 |	0.5 |	0.6 |	0.7 |
> |-------------|-----------|----------|-----------|-----------|
> |LRPCA |	10/10 |	10/10 |	8/10 |	0/10 |
> |ScaledGD |	10/10 |	0/10 |	0/10 |	0/10 |
>
> -----------------------------------------------
>
> ### Question: The section "More numerical experiments" could to with a rewrite. The current text states which additional experiments can be found in the supplements, but it would be better if the text also provided a high-level summary of the findings.
>
> Response: Thank you for this valuable suggestion. **We will follow your suggestion and revise this paragraph in the camera-ready version.**
>
> A high-level summary is listed below:
>
> 1. *Our model has good generalization ability. We can train our model on small-size and low-rank training instances (say, $1000 \times 1000$ and $r=5$) and use the model DIRECTLY on problems with larger size (say, $3000 \times 3000$ and $r=5$) or with higher rank (say, $1000 \times 1000$ and $r=15$) with good performance.*
> 2. *The learned parameters are explainable. With the learned parameters, LRPCA goes very aggressively with large step sizes in the first several steps and tends to be conservative after that.*
>
> -----------------------------------------------
>
> ### Question: I could not find any mention of licenses in Section 4 even if this is stated in the checklist. -- The authors did not include code, but have promised to release upon acceptance. I find this satisfactory.
>
> Response: Thank you for this comment. In the checklist, we stated that “Cannot find license information for the used video dataset”, thus we did not mention any license information in Section 4. We do cite the url link to the used dataset, see footnote 4. If you have any specific instruction on how to process the “no license found” situation, please reply in the followup comment, so we can revise Section 4 accordingly. Thank you in advance.
>
> Regarding the code, we have organized most of the code in the meantime. **We should be able to release the code online as soon as this paper is accepted.**
>
> -----------------------------------------------
> **If you are satisfied with our response, would you kindly bump up your score?**

---

### Official Review · Reviewer_3Wem · 2021-07-14

**Rating:** 6
**Confidence:** 4

**Summary:**

The paper studies the problem of robust PCA. The authors first proposed an improved algorithm based on the previous work ScaledGD. Then they proposed to use "learning to optimize" to select the hyper parameters dynamically and adaptively. The numerical results showed that the proposed method is more effective than two baselines.

**Limitations And Societal Impact:**

Yes.

**Main Review:**

The paper is well-written and the motivation is clearly explained. The proposed LRPCA algorithm (Algorithm 1) is an improved version of the ScaledGD. They are different in the initialization and the update of the sparse matrix S. Theorem 1 can demonstrate the superiority of the proposed algorithm over the ScaledGD. On the other hand, the authors use RNN to determine the step size and thresholding parameter adaptively, which is new to me, though it is still in the traditional framework of "learning to optimize".

The paper has the following weakness.

1. Some statements are not convincing. For example, in lines 34-38, the authors claimed that "Another issue is that the existing approaches only learn the parameters for a finite number of iterations". But I think this is not an issue. Nobody wants to use infinite number of iterations. "Moreover, if the user desire a better accuracy later, one may have to restart the learning process to obtain parameters of the extra iterations". One can just store the iterative information and restart from any iterations.

2. In Theorem 1, the alpha should be no larger than a value scaled by 10^4. It is better to provide some examples of the values of \mu, r, \kappa to show that \alpha can be large relatively.

3. In Figure 3, it seems that the initialization plays a very important role in LRPCA. It is not clear that why the proposed method outperform the ScaledGD when alpha is larger.

4. In Figure 4, it is not clear why LRPCA has less time cost per iteration than the ScaledGD. The authors haven't provided the computational cost of training the RNN. I think the per-iteration cost of the proposed method should be higher than that of the ScaledGD because it involves the RNN.

5. In Table 1, the stop criterion may not be a good choice because the reconstruction error is sensitive the the sparsity of S. The authors may use the relative changes of the variables X and S instead.


**Time Spent Reviewing:**

4

---

> ### Author Response · Authors · 2021-08-10
> **Response to Reviewer 3Wem**
>
> ### Question:  Some statements are not convincing. For example, in lines 34-38, the authors claimed that "Another issue is that the existing approaches only learn the parameters for a finite number of iterations". But I think this is not an issue. Nobody wants to use infinite number of iterations. "Moreover, if the user desire a better accuracy later, one may have to restart the learning process to obtain parameters of the extra iterations". One can just store the iterative information and restart from any iterations.
>
> Response: Thank you for this interesting question. The issue is with the choice of the finite iteration-number. More iterations, thus a deeper network, are difficult to train; and, the difficulty increases fairly quickly with the depth. Fewer iterations limit the convergence quality; if the target test accuracy is not reached by the last iteration, restarting the network from the current output usually does NOT work since the network is not trained to adapt to start from its own results.  Going infinite does not mean actually taking infinitely many iterations but rather the choice to run as many of the iterations as deemed necessary (e.g., until reaching the target accuracy or the time limit) and our network is trained to be used like this.
>
> -----------------------------------------------
>
> ### Question: In Theorem 1, the alpha should be no larger than a value scaled by 10^4. It is better to provide some examples of the values of $\mu$, $r$, $\kappa$ to show that \alpha can be large relatively.
>
> Response: Thank you for this interesting question. In this field of research, people usually care about only the order of $\alpha$ tolerance, and the constant is often hidden/ ignored. As long as $\alpha$ is independent of $n$, then the result is desirable. In contrast, $\alpha \lesssim O(1 / \sqrt{n})$ is considered suboptimal.
>
> The reason we use this constant in the theorem is to match the constant in the analysis of ScaledGD. Thus, we can have an apple-to-apple comparison, i.e., our advantage of allowing larger step sizes is not coming from a smaller constant. That being said, we can prove the theorem with a much smaller constant with shorter step sizes.
>
> For your interest, $\mu \in [1, n/r]$, $r \in [1, n]$, and $\kappa \in [1, \infty)$. Generally speaking, a well-posed RPCA problem has $\mu, r, \kappa = O(1)$ with respect to the problem dimension $n$. We have mentioned the setting of $\mu$ in Assumption 1. **We will add some short discussion for $\kappa$ and $r$ in the camera-ready version.**
>
> -----------------------------------------------
>
> ### Question: In Figure 3, it seems that the initialization plays a very important role in LRPCA. It is not clear that why the proposed method outperform the ScaledGD when alpha is larger.
>
> Response: Thank you for this interesting question. The good initialization in Figure 3 is a result of learning. ScaledGD, with the sparsification operator, cannot achieve similar initialization through hand-tuned parameters.
>
> The thresholding operator (in LRPCA) has two advantages over the sparsification operator (in ScaledGD): speed and no false-positives (which leads to better robustness). The speed advantage is obvious as the sparsification operator requires partial sorting in every row and every column. For robustness advantage, we will use an example. Recall that ScaledGD takes off the largest $\gamma \alpha$ fraction entries per row and per column via the sparsification operator. Let’s take the parameter $\gamma = 1.5$ for now ($\gamma$ must be strictly greater than $1$, and ScaledGD uses $\gamma=2$ for analysis). When we have only 10% outliers, the sparsification operator takes off 15% entries---not too bad.  If we have 40% outliers,  the sparsification operator takes off 60% entries---lose too many good entries. In contrast, the thresholding operator (with a properly chosen thresholding value) in LRPCA takes off no more than 40% entries, and all good entries are saved. In this sense, LRPCA preserves more good redundant information than ScaledGD, especially when more outliers appear. Hence,  it is not a surprise that LRPCA outperforms ScaledGD when $\alpha$ is larger. **We are happy to add a remark in the camera-ready version to clarify any ambiguity in this matter.**
>
> In addition, we add the following experiment to support our claim:
>
> *We generate 10 problems for each of the outlier levels (i.e., $\alpha$) and compare the recoverability of LRPCA and ScaledGD. The test result is presented in the table below, where we can conclude that LRPCA is more robust when $\alpha$ is larger.*
>
> |$\alpha$ |	0.4 |	0.5 |	0.6 |	0.7 |
> |-------------|-----------|----------|-----------|-----------|
> |LRPCA |	10/10 |	10/10 |	8/10 |	0/10 |
> |ScaledGD |	10/10 |	0/10 |	0/10 |	0/10 |
>
> -----------------------------------------------
>
> ### Question: In Figure 4, it is not clear why LRPCA has less time cost per iteration than the ScaledGD. The authors haven't provided the computational cost of training the RNN. I think the per-iteration cost of the proposed method should be higher than that of the ScaledGD because it involves the RNN.
>
> Response: Thank you for this valuable comment. In our approach, we actually do NOT involve an extra RNN in each iteration. We refer to the notion “RNN” because different iterations in our algorithm share the same parameters $\beta$ and $\phi$ and, consequently, the whole loop can be viewed as a RNN. The only extra calculation caused by RNN is equation (13): $\eta_k = \beta \cdot \eta_{k-1}, \zeta_k = \phi \cdot \zeta_{k-1}$. They are just two times of scalar multiplications, which can be ignored compared with the matrix calculations.
>
> -----------------------------------------------
>
> ### Question: In Table 1, the stop criterion may not be a good choice because the reconstruction error is sensitive the sparsity of $S$. The authors may use the relative changes of the variables $X$ and $S$ instead.
>
> Response: Thank you for this valuable suggestion. **We have made this change and will revise the experimental results accordingly in the camera-ready version.**
>
> The new stopping condition is:
> $\max(\frac{||X_k-X_{k-1}||} {||X_{k-1}||}, \frac{||S_k-S_{k-1}||}{ ||S_{k-1}||})< 10^{-3}$,
> where $||\cdot||$ is Frobenius norm (markdown has some display issue, we have to write it in this way here. The standard Frobenius norm notation will be used in the camera-ready version).
>
> For your convenience, we attach the revised Table 1 here:
>
> |Video |	 LRPCA |	Scaled GD |	AltProj |
> |----------------|-----------------|-----------------|-----------------|
> |ParkingLot1 |	 16.01 |	260.45 |	63.04 |
> |ParkingLot2 |	 33.95 |	639.03 |	144.50|
> |ParkingLot3 |	 38.85 |	662.08	|	166.91|
> |StreeView |	 33.73 |	626.05 |	167.66|
>
> The revised visual results will be updated in the revision.
>
>
> -----------------------------------------------
> **If you are satisfied with our response, would you kindly bump up your score?**

---

### Official Review · Reviewer_NqV9 · 2021-07-15

**Rating:** 6
**Confidence:** 3

**Summary:**

The authors study the problem of robust PCA inspired  by a LISTA type approach. In particular, the authors consider  a hybrid model of feedforward, and recurrent NN architectures and "learn the parameters of (scaled) GD" that is used to solve RPCA. I am not extremely familiar with the literature, but essentially the hybrid model allows for a non-apriori (in terms of the architecture) number of iterations that the algorithm can run for. This seems particularly interesting to me. The authors show that under certain conditions of the parameters, the algorithm enjoys linear convergence. The authors also show synthetic data experiments and a use-case of background subtraction to validate their results.

**Limitations And Societal Impact:**

Please see above.

**Main Review:**

Overall, the paper is well written and the thought process is clearly explained. Here are some detailed comments:

**Originality:** A key contribution of this paper seems to be a hybrid Neural Network architecture that exploits best-of-both-worlds of Feedforward NNs and Recurrent NNs. This allows for the iterative optimization algorithm (essentially a modified version of scaled Gradient Descent) for *infintely* many iterations. To my best knowledge, I am not aware of such a approach. Thus, it would be nice to see a slightly more detailed review of such a discussion (on this line of work).


**Quality:** The paper is well written and concepts are explained clearly. I think this is an interesting result, but I have the following main comments/questions:

**Experiments**

  - Firstly, the experiments are not very convincing. The reported run-time is definitely not a fair metric since the proposed method requires a massive NN that needs to be trained. The time taken for such training needs to be reported in ALL experiments. While I understand that training is a one-time cost incurred, this is also technically not true since if the data set changes (in test time, or said another way *if the network encounters an out of distribution sample*) then of course the network needs to be trained again.

  - In the same vein as above, this makes the claim of the "constant factor" improvement in SVD not as impressive. Also to clarify, how do the authors attain this precise per iteration complexity (described in footnote 3)? There is no mention of this contribution anywhere later in the paper.

- The experiments only show a run time comparison with existing methods, what about the output of the background subtraction?  This definitely needs to be included to show it outperforms (or performs at par) with existing methods.

**Theoretical Contributions**

- Firstly, the claim that the algorithm is better than state-of-the-art is misleading. There are key references (such as *Nearly Optimal Robust Matrix Completion*, Cherapanamjeri et al., '16 and references therein) that are missing. These provide better time complexity and outlier-tolerance (under the exact model considered in this paper).

- Second, the claim that the result is better than [10] of the paper is also debatable. While I understand that tolerating a higher step-size is somewhat beneficial, a crtical point that has not been elaborated upon is that the $\zeta_k$'s require the knowledge of $X_{\ast}$ whereas [10] makes no such assumptions.

- The fact that the threshold in soft-thresholding is reducing in every iteration is not a new idea.  This has in fact been the basis for *Non-Convex Robust PCA*, Netrapalli et al., '14, and follow up works


**Clarity:** The paper is well written and explained.

**Significance:** The paper introduces some new ideas regarding a hybrid model of feedforward and recurrent NN architectures. This seems to allow more control in iterative optimization procedures that is instantiated on a RPCA problem.

-----------------------

Overall, I believe that the paper has some good ideas, but the authors need to tone the discussion down  wherever applicable.

**Time Spent Reviewing:**

3

---

> ### Author Response · Authors · 2021-08-10
> **Response to Reviewer NqV9 - Part 2**
>
> ### Question: Firstly, the claim that the algorithm is better than state-of-the-art is misleading. There are key references (such as Nearly Optimal Robust Matrix Completion, Cherapanamjeri et al., '16 and references therein) that are missing. These provide better time complexity and outlier-tolerance (under the exact model considered in this paper).
>
> Response: Thank you for bringing up this key reference. [Cherapanamjeri et al. '16] uses the trick of iterative random subsample to reduce the computational complexity and the recovery is guaranteed *with high probability*, which is smart and impressive. In fact, the similar trick can be applied to factorized gradient descent methods too---GD [Yi et al. ‘16] (cited as [8] in paper) proposes and analyses partially observed RPCA, which is practically equivalent to the subsample idea from [Cherapanamjeri et al. '16]. In our paper, we consider the setting of fully observed RPCA with deterministic approaches. The subsample trick is definitely interesting and is indeed one of our future directions. Keep in mind, this extension is nontrivial for learning. For instance, to balance between efficiency and recoverability, we must learn to optimize the number of samples which is indifferentiable; thus, we need more advanced machine learning techniques beyond back-propagation that was used in this paper. Note that the “best” number of samples can be iteratively changed via learning. **That being said, we are happy to add [Cherapanamjeri et al. '16] and the related references to the section of related work in the camera-ready version.**
>
> -----------------------------------------------
>
> ### Question: Second, the claim that the result is better than [10] of the paper is also debatable. While I understand that tolerating a higher step-size is somewhat beneficial, a crtical point that has not been elaborated upon is that the $\zeta_k$'s require the knowledge of  $X_\star$ whereas [10] makes no such assumptions.
>
> Response: Thank you for this valuable comment. Theorem 1 is a proof of existence---there exists a set of good parameters for LRPCA to outperform ScaleGD. We must clarify that LRPCA does not require the knowledge of $X_\star$. $X_\star$ is only used to construct the theoretical parameters. No one knows exactly what kind of parameters SGD-based learning will get. However, numerical experiments show that we tend to get better parameters than what the theory predicts. Therefore, to a certain extent, given the theoretical parameter selection based on $X_\star$, effective training tends to yield even better parameters. **To your satisfaction, we are happy to add a sentence between Lines 193-195 to stress that the theoretically good parameters rely on the knowledge of $X_\star$, but, again, it is ONLY to construct the theoretical parameters.**
>
> -----------------------------------------------
>
> ### Question: The fact that the threshold in soft-thresholding is reducing in every iteration is not a new idea. This has in fact been the basis for Non-Convex Robust PCA, Netrapalli et al., '14, and follow up works
>
> Thank you for this valuable comment. [Netrapalli et al. ‘14] is cited as AltProj [6] in our paper.
> Firstly, AltProj and its followup works, such as [Cherapanamjeri et al. '16] and [Cai et al. ‘19] (cited as [9] in our paper), use hard-thresholding not soft-thresholding. Although there are many similarities between hard- and soft-thresholdings, they are still very different operators. Secondly, as stated in the last response above, our theorem shows the existence of good parameters. The fact that the “best” threshold values are iteratively reduced is a result of learning. Thirdly, analyzing the thresholding operator within the factorized gradient descent framework is indeed challenging and involved---our theoretical result is NOT a trivial combination of AltProj and ScaledGD by all means. To our best knowledge, such analysis has never been done before. **We believe the theorems in AltProj do not diminish our theoretical contribution. Nevertheless, we are happy to add sentences at Lines 59 and 141 such that AltProj and its followup works get the credits they deserve.**
>
> -----------------------------------------------
> **If you are satisfied with our response, would you kindly bump up your score?**

---

> ### Author Response · Authors · 2021-08-10
> **Response to Reviewer NqV9 - Part 1**
>
> ### Question: A key contribution of this paper seems to be a hybrid Neural Network architecture that exploits best-of-both-worlds of Feedforward NNs and Recurrent NNs. This allows for the iterative optimization algorithm (essentially a modified version of scaled Gradient Descent) for infinitely many iterations. To my best knowledge, I am not aware of such an approach. Thus, it would be nice to see a slightly more detailed review of such a discussion (on this line of work).
>
> Response: Thank you for your valuable comment. To develop good learning-based optimization algorithms, researchers usually parameterize an existing optimization algorithm and learn its parameters. In a perspective of parameterization approaches, these works can be divided into two categories:
> 1. Feedforward neural networks: This approach dates back to [Gregor & LeCun. ‘10] (cited as [19] in our paper). It unfolds ISTA, an iterative algorithm for LASSO, and truncates it into finite steps. The parameterized ISTA can be viewed as a FNN, thus the parameters can be learned via backpropagation. Its following works [Yang et al. ‘16, Zhang & Ghanem. ‘18] achieve great success on signal related problems, such as compressed MRI.
> 2. Recurrent neural networks: This approach was first proposed in [Andrychowicz et al. ‘16] and gains increasing attention in the area of “learning to learn” [Wichrowska et al. ‘17, Metz et al. ‘19]. Since many machine learning algorithms, such as SGD and Adam, need a large number (say, 5k~10k) of iterations to stop. It is hard to follow the above unfolding idea directly due to the long trial. Thus, a RNN is used to parameterize such optimization algorithms. Since RNN layers share common parameters, memory cost is significantly reduced.
>
> To the best of our knowledge, our work is the first one to utilize the advantages of both FNN and RNN---using FNN to learn the parameters in significant iterations and using RNN to represent the rest iterations. **We will add the above discussion to the section of related work in the camera-ready version.**
>
> New Reference:
>
> [Yang et al. ‘16] Yang, et al. "Deep ADMM-Net for Compressive Sensing MRI." NeurIPS, 2016.
>
> [Zhang & Ghanem. ‘18] Zhang and Ghanem. "ISTA-Net: Interpretable optimization-inspired deep network for image compressive sensing." CVPR, 2018.
>
> [Andrychowicz et al. ‘16] Andrychowicz, et al. "Learning to learn by gradient descent by gradient descent." NeurIPS, 2016.
>
> [Wichrowska et al. ‘17] Wichrowska, et al. "Learned optimizers that scale and generalize." ICML, 2017.
>
> [Metz et al. ‘19] Metz, et al. "Understanding and correcting pathologies in the training of learned optimizers." ICML, 2019.
>
> -----------------------------------------------
>
> ### Question: Firstly, the experiments are not very convincing. The reported run-time is definitely not a fair metric since the proposed method requires a massive NN that needs to be trained. The time taken for such training needs to be reported in ALL experiments. While I understand that training is a one-time cost incurred, this is also technically not true since if the data set changes (in test time, or said another way if the network encounters an out of distribution sample) then of course the network needs to be trained again.
>
> Response: Thank you for your valuable comment. Given a particular application environment (i.e., a class of similar problems), it is a matter of how the training dataset is generated. With properly selected training examples, the training is indeed a one-time cost. If the out-of-distribution samples frequently appear, it probably indicates a badly generated training dataset. By the way, when the problems belong to the same class, LRPCA has very good generalization ability, in terms of different $n$ and/or $r$ (see Section B.2).
> With a new application, of course, LRPCA will have to be retrained---just like all other learning-based methods.
> Our training time with different matrix sizes, ranks, and outlier ratios are reported in the following tables:
>
> |Problem settings	|	Training time|
> |----------------------------|--------------|
> |$n=1000, r=5, \alpha=0.1$	|	$1208$ secs|
> |$n=1000, r=5, \alpha=0.2$	|	$1209$ secs|
> |$n=1000, r=5, \alpha=0.3$	|	$1208$ secs|
>
>
>
> |Problem settings	|	Training time|
> |----------------------------|--------------|
> |$n=1000, r=5, \alpha=0.1$	|	$1208$ secs|
> |$n=3000, r=5, \alpha=0.1$	|	$1615$ secs|
> |$n=5000, r=5, \alpha=0.1$	|	$2405$ secs|
>
>
>
> |Problem settings	|	Training time|
> |----------------------------|--------------|
> |$n=1000, r=5, \alpha=0.1$	|	$1208$ secs|
> |$n=1000, r=10, \alpha=0.1$|	$1236$ secs|
> |$n=1000, r=15, \alpha=0.1$|	$1249$ secs|
>
> Different from the inference time reported in our paper, the training was done on a workstation equipped with two GeForce RTX 3080 GPUs. Note that the training time is not proportional to the problem size due to the high concurrency of GPU computing. **We will add the above tables to the camera-ready version.**
>
> -----------------------------------------------
>
> ### Question: Also to clarify, how do the authors attain this precise per iteration complexity (described in footnote 3)? There is no mention of this contribution anywhere later in the paper.
>
> Response: Thank you for pointing this out. Here is the breakdown of LRPCA’s complexity:
> 1.   Compute $L_k R_k^T$: $n$-by-$r$ matrix times $r$-by-$n$ matrix, $n^2 r$ flops.
> 2.   Compute $Y - L_k R_k^T$: $n$-by-$n$ matrix minus $n$-by-$n$ matrix, $n^2$ flops.
> 3.   Soft-thresholding on $Y - L_k R_k^T$: one pass on $n$-by-$n$ matrix, $n^2$ flops.
> 4.   Compute $L_k R_k^T + S_{k+1} - Y = S_{k+1} - (Y - L_k R_k^T)$: $n$-by-$n$ matrix minus $n$-by-$n$ matrix, $n^2$ flops.
> 5.   Compute $R_k^T R_k$: $r$-by-$n$ matrix times $n$-by-$r$ matrix, $n r^2$ flops.
> 6.   Compute  $(R_k^T R_k)^{-1}$: invert a $r$-by-$r$ matrix, $O(r^3)$ flops.
> 7.   Compute $R_k (R_k^T R_k)^{-1}$: $n$-by-$r$ matrix times $r$-by-$r$ matrix, $n r^2$ flops.
> 8.   Compute $(L_k R_k^T + S_{k+1} - Y) \cdot  R_k (R_k^T R_k)^{-1}$: $n$-by-$n$ matrix times $n$-by-$r$ matrix, $n^2 r$ flops.
> 9.   Compute $L_{k+1}= L_k - \zeta_{k+1} (L_k R_k^T + S_{k+1} - Y)  R_k (R_k^T R_k)^{-1}$: $n$-by-$r$ matrix minus scalar times $n$-by-$r$ matrix, $2nr$ flops.
> 10. Repeat step 5 - 9 for computing $R_{k+1}$: another $2n r^2 + O(r^3) + n^2 r +2n r$ flops.
>
>
> In total, LRPCA requires $3 n^2 r + 3 n^2 + O(n r^2)$ flops per iteration provided $r \ll n$.  Note that we count $abc$ flops for computing an $a$-by-$b$ matrix times a $b$-by-$c$ matrix in the above complexity calculation. Some may argue that this matrix product should take $2abc$ flops. We can rectify the per-iteration complexity to $6 n^2 r + 3 n^2 + O(n r^2)$ flops if the reviewers prefer the latter opinion. **We realize this detail was missed in the current submission and will summarize the above calculation to the section of proposed method (after Line 158) in the camera-ready version**.
>
> -----------------------------------------------
>
> ### Question: The experiments only show a run time comparison with existing methods, what about the output of the background subtraction? This definitely needs to be included to show it outperforms (or performs at par) with existing methods.
>
> Response: Thank you for pointing this out. We have those pictures you would like to see, but we cannot include them in this submission due to the page limit. **We will include the visual outputs of other compared algorithms in the camera-ready version where we should have an extra page** (if there is no extra page this year, we will have to put them in the appendix).

---

### Official Review · Reviewer_P4S4 · 2021-07-22

**Rating:** 7
**Confidence:** 3

**Summary:**

This paper presents a a new approach to solve the Robust PCA problem. They use matrix factorisation to formulate RPCA as a more scalable problem, then learn (or optimise) parameters during the algorithm's each iteration. They also add a recurrent layer at the end to allow for "unlimited" iterations. The methods is compared against classic RPCA solvers, but not against learned methods.

**Limitations And Societal Impact:**

I don't see any obvious limitations or societal impact

**Main Review:**

Overall the paper is well written and well motivated. The proposed method makes a huge improvement compared to the classical solvers.
My main criticism is that the authors don't compare against other learned methods arguing that they are specialised on problem types. However, for a practitioner it doesn't matter if the method is generic for their specific problem, they just need the best one. Hence, I'd encourage the authors to show performance results against specialised learned methods. Even if each specialised method is marginally better than this generic one on one problem, this methods is still a major contribution to the scientific community. However, I think it is important to understand how they compare for each setting.

**Time Spent Reviewing:**

2 hours

---

> ### Author Response · Authors · 2021-08-10
> **Response to Reviewer P4S4**
>
> ### Question: Overall the paper is well written and well motivated. The proposed method makes a huge improvement compared to the classical solvers. My main criticism is that the authors don't compare against other learned methods arguing that they are specialised on problem types. However, for a practitioner it doesn't matter if the method is generic for their specific problem, they just need the best one. Hence, I'd encourage the authors to show performance results against specialised learned methods. Even if each specialised method is marginally better than this generic one on one problem, this methods is still a major contribution to the scientific community. However, I think it is important to understand how they compare for each setting.
>
>
> Response: We appreciate your encouragement. There were two reasons why we did not compare with the other learned methods: 1. As you mentioned, they are specialised.  2. They involve the costly full SVD every iteration, which makes their training cost extremely high for large-dimension problems. At your request, we completed the comparisons. Here is the result:
>
> We compare LRPCA with CORONA [20], a state-of-the-art learning-based RPCA method, on ultrasound imaging. Their simulation experiment data, consisting of $2400$ training examples and $800$ validation examples, can be downloaded online. Each example is of size $1024 \times 40$. We compared our model with their trained network on the validation set and we set the underlying rank of the low-rank matrix as $r=1$. Recovery accuracy is measured by a loss function defined in [20]:
> $Loss = MSE(X_{output}, X_\star)$,
> where MSE stands for the mean square error. The test results are summarized in the following table:
>
>
> | Algorithm   | Average inference time | Average loss |
> |-------------|----------------------------|--------------|
> | CORONA [20] | 0.9225 secs                | 4.9e-4       |
> | LRPCA       | 0.0057 secs                | 9.9e-4       |
>
> Note that LRPCA is designed for generic RPCA and CORONA is specifically designed for ultrasound imaging. Thus, it is not a surprise that our recovery accuracy is slightly worse. However, our runtime is substantially faster than CORONA. We believe that our speed advantage will be even more significant on larger examples, which is indeed our main contribution: scalability.
>
> **In the camera-ready version, we are happy to include the above experiment.**
>
> -----------------------------------------------
> **If you are satisfied with our response, would you kindly bump up your score?**

---

### Author Response · Authors · 2021-08-26
**Did we address your questions and concerns? Your feedback would be appreciated**

To all reviewers:

We really appreciate all the reviewers for their valuable suggestions. We sincerely hope to have further discussion with the reviewers to see if our response solves their concerns. We are confident that our response should have cleared the air, and we can clarify more if there is more need. We are happy to answer any additional questions and provide more information.

Paper 8361 Authors

---

> ### Comment · Reviewer_3Wem · 2021-08-31
> **Thanks for the response. I keep may score.**
>
> Hi Authors,
>
> Thanks a lot for the detailed response. The work provided some interesting results. However in my opinion, the contribution to PCA or optimization is not significant enough. I will not change my score.
>
> Reviewer

---

### Decision · Program_Chairs · 2021-09-27

**Decision:**

Accept (Poster)

**Comment:**

The paper studies learned approaches to the problem of recovering a low-rank matrix from sparsely corrupted observations (here, RPCA). This problem has been intensely studied, and a number of methods have been proposed based on both convex and nonconvex optimization. The paper takes the perspective of *learned* optimization, in which one learns parameters of an optimization method from data, by interpreting iterations of the method as layers of a neural network. This has the advantage of adapting the method to (1) perform as well as possible for a given (small) number of iterations and (2) to perform as well as possible on data whose distribution mimics that of the training data. There are applications of learned optimization to RPCA. However, these approaches are based on the singular value decomposition, and so are not scalable to very large input matrices.

The paper proposes a learned optimization method which combines soft-thresholding / proximal methods and a factorized low-rank model which is optimized using a scaled gradient descent (ScaledGD) which better copes with the ambiguity between factors. The paper uses learning to determine a sequence of step sizes and a sequence of thresholding parameters. The paper allows the learned method to be applied for arbitrary numbers of steps, by treating the layers as a recurrent neural net which can be iterated indefinitely.

The main potential advantages compared to previous learned RPCA approaches are (1) avoiding the need to perform singular value decomposition, (2) the use of RNN’s enables the learned method to be applied for arbitrary numbers of iterations.

The paper provides theoretical analysis showing that there exist parameters which enable linear convergence from an appropriate initialization (given by performing a single truncated SVD, as in many prior works). Experiments show significantly improved convergence speed compared to ScaledGD and alternating minimization.

Initial reviews praised the clear motivation and presentation of the paper, as well as the idea of combining feedforward and recurrent networks to learn more flexible optimizers. At the same time, reviewers raised questions about the novelty of the paper’s proposals, the relationship to other notions of RPCA, and various aspects of the experiments.

After responses from the authors, the reviewers converged to an evaluation of the paper as above the bar for acceptance, appreciating the paper’s empirical improvements, and the idea of combining feedforward and recurrent networks to handle arbitrary numbers of iterations. The paper would benefit from clarifications as to the novelty of the theoretical recovery results, and where technical innovations were required in the proofs.